# SP-VLA: A Joint Model Scheduling and Token Pruning Approach for VLA Model Acceleration

Ye Li[1], Yuan Meng[1]*, Zewen Sun[1†], Kangye Ji[1], Chen Tang[2], Jiajun Fan[3], Xinzhu Ma[2], Shutao Xia[1], Zhi Wang[1]*, Wenwu Zhu[1]
[1] Tsinghua University    [2] MMLab, The Chinese University of Hong Kong
[3] University of Illinois at Urbana-Champaign

## Abstract

Vision-Language-Action (VLA) models have attracted increasing attention for their strong control capabilities. However, their high computational cost and low execution frequency hinder their suitability for real-time tasks such as robotic manipulation and autonomous navigation. Existing VLA acceleration methods primarily focus on structural optimization, overlooking the fact that these models operate in sequential decision-making environments. As a result, temporal redundancy in sequential action generation and spatial redundancy in visual input remain unaddressed. To this end, we propose **SP-VLA**, a unified framework that accelerates VLA models by jointly scheduling models and pruning tokens. Specifically, we design an action-aware model scheduling mechanism that reduces temporal redundancy by dynamically switching between VLA model and a lightweight generator. Inspired by the human motion pattern of focusing on key decision points while relying on intuition for other actions, we categorize VLA actions into *deliberative* and *intuitive*, assigning the former to the VLA model and the latter to the lightweight generator, enabling frequency-adaptive execution through collaborative model scheduling. To address spatial redundancy, we further develop a spatio-semantic dual-aware token pruning method. Tokens are classified into *spatial* and *semantic* types and pruned based on their dual-aware importance to accelerate VLA inference. These two mechanisms work jointly to guide the VLA in focusing on critical actions and salient visual information, achieving effective acceleration while maintaining high accuracy. Extensive experiments show that our method achieves $1.5\times$ lossless acceleration in LIBERO and $2.4\times$ in SimplerEnv, with up to 6% average performance gain. Inference frequency and latency improve by $2.2\times$ in SimplerEnv and $1.4\times$ in LIBERO. Moreover, on real-robot evaluations, our approach maintains accuracy with only a 1% drop while delivering a $2.5\times$ end-to-end acceleration. The code is available at https://github.com/ChildTang/SP-VLA.

## 1 Introduction

Vision-Language-Action (VLA) models integrate visual perception and language understanding to generate actionable outputs for robotic control and task execution in embodied agents, demonstrating remarkable progress across a wide range of tasks (Zhang et al., 2024; Han et al., 2024; Shi et al., 2025; Figure AI, 2024; Team et al., 2024; Liu et al., 2024a;b). However, VLA models are generally large-scale. For example, Google's recently released RT-X series (Brohan et al., 2023; Belkhale et al., 2024; O'Neill et al., 2024b) contains more than 55 billion parameters, and even lightweight models widely used like OpenVLA (Kim et al., 2024) still exceed 7 billion parameters. The resulting computational burden leads to slow inference, making them unsuitable for real-time scenarios such as industrial control, autonomous navigation, and medical robotics.

---

*Corresponding Authors: yuanmeng@tsinghua.edu.cn, wangzhi@sz.tsinghua.edu.cn. † Work done as research intern at Tsinghua University.

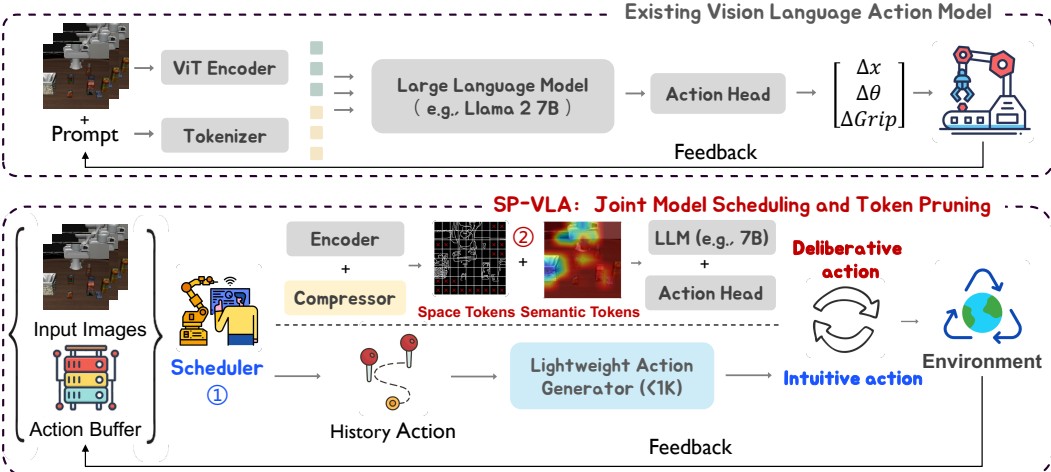

Figure 1: **The main idea of SP-VLA.** Unlike traditional VLA models, SP-VLA first determines the type of the current action. ① For intuitive actions, a lightweight action generator is employed to approximate the output, while for deliberative actions, the high-precision VLA model is used to ensure accuracy. ② When the VLA model is invoked, we further accelerate inference by adaptively pruning tokens based on integrated spatial and semantic information. By jointly leveraging the above two strategies, SP-VLA effectively directs the model's attention to critical actions and salient visual information, achieving substantial speedup without compromising accuracy. Among these components, the action head takes different forms across VLA architectures, such as a D-tokenizer in OpenVLA or a diffusion-based policy in CogACT.

Existing methods for VLA model acceleration focus solely on reducing the single-step computation redundancy via model compression techniques (*e.g.*, pruning (Li et al., 2025b; Kong et al., 2025), quantization (Tang et al., 2024; 2022a;b), speculative decoding (Li et al., 2025a), caching (Li et al., 2024b; Wimbauer et al., 2024; Ma et al., 2024)). Specifically, DeeR-VLA (Yue et al., 2024) introduces an early-exit mechanism that reduces the computational burden of the LLM backbone. QAIL (Park et al., 2024) incorporates an imitation learning mechanism to quantize the VLA model to 4-bit precision, reducing computational cost while preserving model accuracy. VLA-Cache (Xu et al., 2025) reduces computation by selectively reusing tokens deemed less informative. Although these works achieve certain levels of speedup, they primarily focus on accelerating Vision-Language Model (VLM) architectures, overlooking the unique characteristics of VLA models, which introduce an additional temporal dimension by generating actions step-by-step through continuous interaction with the environment. As a result, accelerating VLA models presents two major challenges: (1) Given the temporal dependencies in embodied tasks, how can we effectively leverage historical information to support current decision-making and reduce computational redundancy? (2) Given the high redundancy in visual input from the camera, how can we effectively retain informative visual content and reduce redundancy along the spatial dimension? To this end, we study the problem of handling temporal and spatial redundancy of VLA models for the first time, and present **SP-VLA**, a unified framework that jointly **S**cheduling the model and **P**runing tokens to accelerate **VLA** models.

To solve temporal dependencies, we first design a *action-type aware model scheduling* approach that enables frequency-adaptive inference by dynamically switching between the large-scale VLA model and a lightweight model at different time steps. By rethinking human motion patterns, we observe that deliberate thinking typically occurs only at critical moments such as grasping or turning, while movements between those key points are executed intuitively (Schwartz, 2016; Merel et al., 2019; Murray & Escola, 2020). This allows humans to perform complex tasks both quickly and accurately. Interestingly, we find that **VLA models follow a similar behavioral pattern like human, with actions falling into two categories: *deliberative* and *intuitive*.** Based on this observation, Based on this, we propose using the VLA model to generate *deliberative* actions, while delegating *intuitive* actions to a lightweight model. Specifically, we assume that high-speed movements are typically *intuitive*, whereas low-speed movements are more likely to be *deliberative*, and we determine the action type at each time step based on historical information. To model *intuitive* actions, we exploit

the inertia in object motion and design a lightweight action generator based on Ridge Regression, enhanced with an action cache for efficient prediction. However, since embodied tasks are inherently more complex than simple linear movements, the generation of *intuitive* actions still benefits from the VLA model's directional guidance to ensure task fidelity. As a result, *intuitive* actions are actually produced through high-frequency switching between the lightweight generator and the VLA model.

To mitigate spatial redundancy, we design a *spatio-semantic dual-aware token pruning* to preserve the most relevant visual information. Unlike VLMs, VLA models must understand the relative positions of objects to complete tasks, implying the need for spatial perception. Experimental results show that disrupting token arrangement or aggressively pruning background tokens, such as object contours, leads to significant performance degradation in VLA tasks. This indicates that **the spatial perception of VLA models relies on the relative order of tokens and object contour information.** To address this, we integrate edge and semantic information by extracting object contours using the Canny operator and estimating semantic importance via accumulated attention scores. We also dynamically adjust the token pruning threshold based on current motion speed to maximize inference efficiency.

Overall, we first perform adaptive scheduling between the VLA model and a lightweight action generator based on the action type at each time step. When invoking the VLA model, we further apply speed-aware token pruning, enabling task-aware and frequency-adaptive inference. Through this joint optimization, we effectively eliminate both temporal and spatial redundancies, guiding the model to focus on critical actions and salient visual elements to maximize inference efficiency. Extensive experiments demonstrate that our method achieves $1.5\times$ lossless speedup in LIBERO and $2.4\times$ in SimplerEnv, with up to 6% performance gain, while improving inference frequency and latency by $1.4\times$ and $2.2\times$, respectively. On real-robot evaluations, our approach maintains accuracy with only 1% drop while delivering a $2.5\times$ end-to-end acceleration, highlighting its strong efficiency and robustness.

The main contributions of SP-VLA are as follows:

(1) To the best of our knowledge, this is the first work to accelerate VLA models via reducing the temporal and spatial redundancy. We propose a framework that jointly performs model scheduling and token pruning, guiding the VLA model to focus on key actions and visual elements to achieve maximal acceleration.

(2) We propose an action-aware model scheduling algorithm that effectively reduces temporal redundancy in VLA models. We observe that the VLA action sequences resemble human behavior, comprising *intuitive* and *deliberative* actions. Leveraging this structure, we assign intuitive actions to a lightweight model and deliberative ones to the VLA model, enabling lossless, frequency-adaptive acceleration.

(3) We propose a spatio-semantic dual-aware token compression method. We find that the spatial perception of VLA models relies on the relative positions of tokens and object contour information. Based on this, we design a token pruning method that incorporates both edge features and semantic importance, effectively reducing spatial redundancy.

## 2 RELATED WORK

**Vision-Language-Action Models.** As LLMs gain stronger reasoning ability, the VLA paradigm emerges to extend VLMs to embodied control. DeepMind's RT series, including RT-1 (Brohan et al., 2022), RT-2 (Brohan et al., 2023), RT-X (O'Neill et al., 2024b), and RT-H (Belkhale et al., 2024), are among the earliest large-scale VLA models. Additionally, the release of the Open X-Embodiment (O'Neill et al., 2024a) dataset has laid a strong foundation for continued research. Built on it, OpenVLA (Kim et al., 2024) leverages the reasoning capabilities of LLaMA 2, achieving significant improvements in accuracy. On the other hand, generative models such as diffusion models have been adopted in embodied tasks to enhance the temporal coherence of actions. $\pi_0$ (Black et al., 2024) and $\pi_{0.5}$ (Intelligence et al., 2025) incorporate Flow Matching models as action decoders, allowing VLA models to generate entire action sequences in a single pass, significantly improving execution smoothness and efficiency. Nevertheless, compared to traditional control methods (Li et al., 2023; Liu et al., 2024c; Lin et al., 2022; Li et al., 2021), the lack of attention to action efficiency hampers their deployment in more demanding applications, such as industrial assembly.

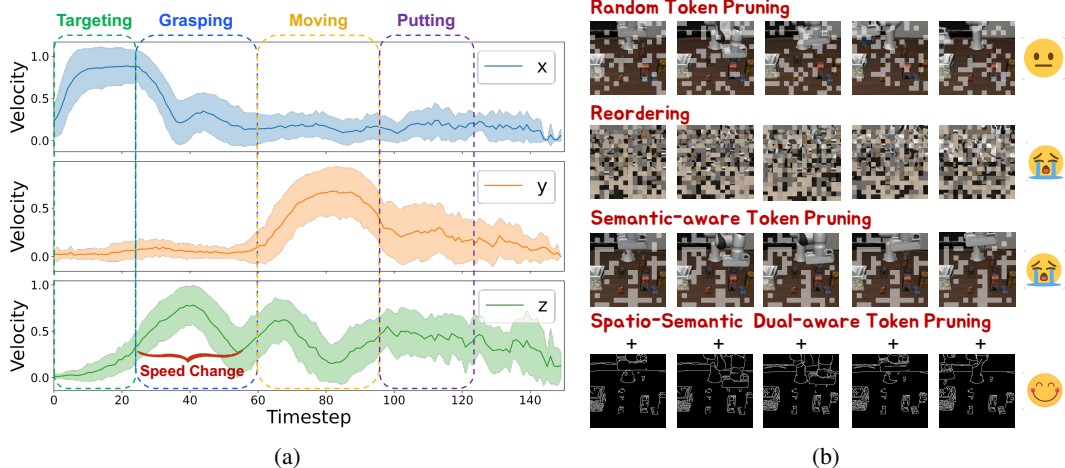

Figure 2: **The visualization of VLA model behavior. (a)** shows the velocity profile of the robot arm across 50 pick-and-place trials, following a consistent four-phase pattern: targeting, grasping, moving, and placing. The VLA model demonstrates complex behavior by adjusting orientation at key points and learning kinematic patterns such as acceleration and deceleration. These action sequences comprise both *deliberative* and *intuitive* components. **(b)** shows task performance under different token distributions. Random pruning degrades accuracy, highlighting the presence of token redundancy. However, relying exclusively on semantic importance, such as through reordering or semantic-aware pruning, causes the model to fail in completing the task. In contrast, integrating *spatial* and *semantic* information enables efficient pruning while preserving performance, as the VLA model relies on token relative positions and object contours for spatial understanding.

**Acceleration for Vision-Language-Action Models.** A lot of work has been devoted to improving the efficiency of the VLA model. QAIL (Park et al., 2024) integrates quantization into the imitation learning fine-tuning process to train quantized policies that approximate expert behavior. Fast (Pertsch et al., 2025) transforms actions into the frequency domain, enabling efficient compression by analyzing their spectral characteristics. Cache methods (Xu et al., 2025; Ji et al., 2025; 2026) accelerates inference by distinguishing background tokens from task-relevant ones and caching the less critical parts. PD-VLA (Song et al., 2025) and VLA-OFT (Kim et al., 2025) modify the autoregressive action generation in VLA models by introducing parallel decoding, significantly improving generation efficiency. However, these approaches fail to account for the specific nature of embodied tasks, such as leveraging historical information and addressing visual redundancy, thus leaving significant potential for further acceleration. Our approach adaptively switches between the VLA model and a lightweight action generator based on *intuitive* and *deliberative* actions, and prunes tokens according to task complexity, enabling frequency-adaptive acceleration.

## 3 A JOINT MODEL SCHEDULING AND TOKEN PRUNING APPROACH FOR VLA MODEL ACCELERATION

In this section, we provide a detailed introduction to SP-VLA. The framework is shown in Fig. 1. Prior to processing environmental feedback, the historical action sequence is analyzed to determine whether the current step requires a *deliberative* or *intuitive* action. *Intuitive* actions are generated using a lightweight generator, while *deliberative* actions are handled by the VLA. Furthermore, before entering the LLM backbone, input tokens are pruned based on their *spatial* context and *semantic* importance, further reducing computational overhead. The lightweight action generator will be introduced in Section 3.1, and the token pruning strategy will be detailed in Section 3.2.

### 3.1 ACTION TYPE-AWARE MODEL SCHEDULING

Human motor behavior relies on deliberate thinking only for complex actions, such as grasping or turning, while other simple actions are executed intuitively (Schwartz, 2016; Merel et al., 2019;

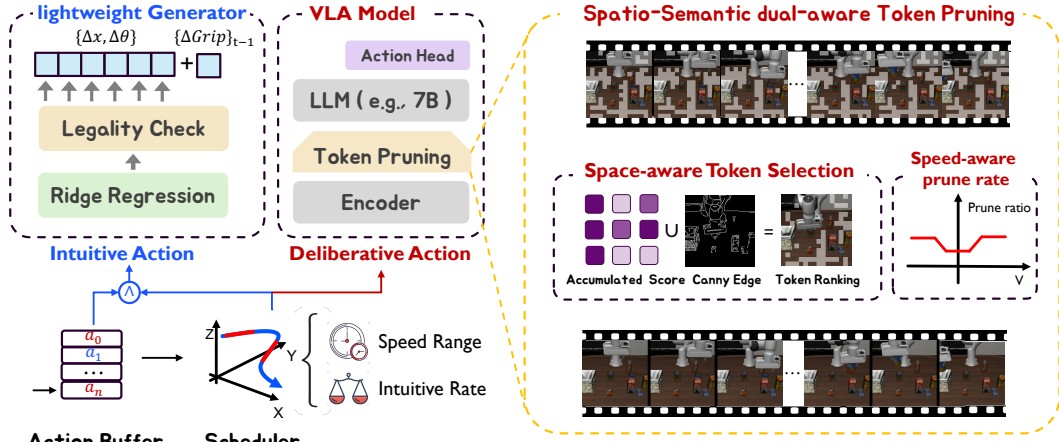

Figure 3: **The framework of SP-VLA.** SP-VLA accelerates the inference process through joint model scheduling and token pruning. ***Left:*** At each time step $t$, the scheduler classifies the current action as *intuitive* or *deliberative* based on the historial trajectories in the action buffer. For intuitive actions, Ridge Regression estimates the translational and rotational components, reusing the gripper state at $t-1$. Otherwise, the VLA model will generate a fine-grained action. ***Right:*** To support spatial understanding, we rank token importance by combining *spatial* information from the Canny operator with *semantic* importance, and perform velocity-adaptive pruning for optimal acceleration.

Murray & Escola, 2020). This hybrid strategy achieves high efficiency and low energy consumption without sacrificing effectiveness. However, existing VLA models treat all actions as equally important, relying on large model (*e.*g., parameter $> 7B$) to generate each action through complex reasoning. In reality, coherent action sequences involve not only high-level logical reasoning but also low-level physical dynamics, including inertia and linear acceleration or deceleration during point-to-point movements, which poses a significant challenge for VLA modeling. Ignoring the distinction between action types leads to substantial redundant computation and ultimately compromises motion smoothness. Therefore, leveraging this property to reduce the computational burden of VLA models is a pressing challenge that needs to be addressed.

**Action Type Indicator.** In order to identify *intuitive* actions in VLA-generated trajectories, we analyzed the behavioral patterns of VLA models and uncovered consistent patterns in grasping tasks. The robotic arm typically performs a slow alignment with the target, then approaches the target position at high speed, and finally executes the grasping action at a moderately low speed. A similar motion pattern is observed during the placement phase, and the observed behavior is illustrated in Fig. 2a. As shown in the figure, the VLA model not only learns logical reasoning capabilities but also captures dynamic patterns such as acceleration and deceleration. Therefore, we conclude that *deliberative* actions are required for precise operations such as turning and grasping, whereas *intuitive* actions are more appropriate during high-speed transitions between task phases. In this paper, we treat the action output of the VLA model as a displacement per time step, *i.*e., velocity. Let $\mathbf{a}_{t_d} = \{a_x, a_y, a_z\}$ represent the translational velocity components of the end-effector at time step $t$. An action $\mathbf{a}_{in} \in \{\mathbf{a} \in \mathbb{R}^l \mid |a_i| > v_{\min}, \forall i \in \{x, y, z\}\}$ is classified as an *intuitive* action if all components exceed a predefined threshold $v_{\min}$, otherwise, it will be considered as a *deliberative* action.

**Model Schedular.** Based on the above conclusion, we determine whether to use the lightweight model based on motion speed and the action cache, as shown in Fig. 3. Low velocities typically indicate fine manipulation, whereas high velocities increase the risk of significant errors when relying on the lightweight model. If $\mathbf{a}_{t-1} \in \{\mathbf{a} \in \mathbb{R}^l \mid v_{\min} < |a_i| < v_{\max}, \forall i \in \{x, y, z\}\}$, the lightweight model can be called, $v_{\min}$ and $v_{\max}$ denote the velocity thresholds. On the other hand, we also monitor the number of VLA-generated actions $N_G$ in the action buffer $\mathbf{S_A}$, and allow the lightweight model to be used when $N_G/N_A > \tau$, where $N_A$ is the total action number of $\mathbf{S_A}$, $\tau$ is a predefined

threshold. Overall, the triggering conditions for the lightweight model are:

$$\text{LWM} = \begin{cases} 1, & \text{if } \mathbf{a}_{t-1} \in [v_{\min}, v_{\max}] \text{ and } \frac{N_G}{N_A} > \tau \\ 0, & \text{otherwise} \end{cases}. \tag{1}$$

By performing small-step, high-frequency model switching, we can achieve faster inference while maintaining the accuracy of action direction.

**Lightweight Action Generator.** To support fast and reliable action approximation, we develop a lightweight generator using Ridge Regression and an action buffer to efficiently estimate upcoming actions. Although the end-effector trajectory of the manipulator is complex, we assume that short segments of *intuitive* actions can be approximated as linear. Therefore, by modeling the relationship between time and velocity in the action buffer, the current action $\mathbf{a}_t$ can be predicted. Specifically, the action buffer $\mathbf{S_A} = \{\mathbf{a}_{t-n}, \mathbf{a}_{t-n+1}, \cdots, \mathbf{a}_{t-1}\}$ is used to store actions generated over the most recent $n$ steps, $t$ is the current timesteps, $\mathbf{a}_t = \{a_1, a_2, \cdots, a_l\}$ is the $l$-dimensional action vector at $t$. $\mathbf{T} = [0, 1, \cdots, n-1]^T$ is the timestep vector. The formulation of the Ridge Regression model is $\mathbf{Y} = \mathbf{X}\boldsymbol{\beta} + \varepsilon$, where $\mathbf{X} = [\mathbf{T}, \mathbf{1}] \in \mathbb{R}^{n \times 2}$ is the input, $\boldsymbol{\beta} \in \mathbb{R}^{2 \times l}$ denotes the parameter matrix to be fitted, $\varepsilon$ is the error term, $\mathbf{Y} \in \mathbb{R}^{n \times l}$ is the action buffer. To generate each new actions, the model is re-fitted from scratch, with the following loss function:

$$J(\boldsymbol{\beta}) = ||\mathbf{X}\boldsymbol{\beta} - \mathbf{Y}||^2 + \lambda||\boldsymbol{\beta}||^2, \tag{2}$$

where $||\boldsymbol{\beta}||^2$ is the Tikhonov regularization term, which imposes an L2 penalty on the parameters, $\lambda$ is the regularization term. The analytical solution to this equation is given by:

$$\boldsymbol{\beta} = (\mathbf{X}^T\mathbf{X} + \lambda\mathbf{I})^{-1}\mathbf{X}^T\mathbf{Y}, \tag{3}$$

where $\mathbf{I} \in \mathbb{R}^{2 \times 2}$ is the identity matrix. Once the optimal parameters of the current segment $\boldsymbol{\beta}^*$ is obtained, the action at the current time step can be calculated as follows:

$$\mathbf{a}_t = \mathbf{x}_t\boldsymbol{\beta}^*, \quad \text{where } \mathbf{x}_t = [t \quad 1]. \tag{4}$$

It is worth noting that, since the end-effector state in this work is represented as a binary variable, we do not apply the above fitting strategy. Instead, we directly reuse the value from the previous time step $t-1$, and delegate state transitions of the end-effector to the VLA model. Finally, the predicted action is directly executed after passing a validity check.

## 3.2 SPATIAL-SEMANTIC DUAL-AWARE TOKEN PRUNING

To further reduce computation, we adopt a data-centric perspective and dynamically prune less important tokens during VLA invocation, enabling the model to concentrate its attention on task relevant content. Since the LLM accounts for the majority of computational overhead in VLA models, we perform token pruning before feeding tokens into the LLM, ensuring compatibility with diverse VLA architectures. Notably, we observe that VLA models are highly sensitive to both the relative positions of input tokens and object contour-related tokens, as evidenced by the experimental results in Fig. 2b. As illustrated, randomly dropping tokens reduces accuracy but does not prevent task completion, suggesting that many tokens are redundant. It is worth noting that even without pruning, reordering tokens solely according to their semantic importance results in task failure, underscoring the importance of token relative positioning for spatial understanding in VLA models. Moreover, even without altering the relative positions of tokens, pruning solely based on semantic importance can remove critical background information, also leading to task failure. Finally, reintroducing positional tokens restores model performance, underscoring the critical role of both token relative ordering and object contour-related tokens in supporting accurate spatial localization.

**Semantic-aware Token Importance.** Given the input image $\mathbf{X}$, the vision encoder transforms it into a sequence of tokens. We use the final layer of the encoder as the basis for token selection. The queries, keys, and values can be calculated as follows:

$$\mathbf{Q} = \mathbf{X}\mathbf{W}_q, \ \mathbf{K} = \mathbf{X}\mathbf{W}_k, \ \mathbf{V} = \mathbf{X}\mathbf{W}_v, \tag{5}$$

Table 1: **Comparisons with the state-of-the-arts on LIBERO.** To adapt these methods for the VLA model, we introduced several enhancements. Specifically, "+R" denotes the preservation of relative token positions, while "+S" represents the incorporation of Canny edge information. Compared to existing approaches, our method achieves a $1.35\times$ speedup without any performance loss. Furthermore, it enables a $1.5\times$ acceleration with less than a 3% drop in accuracy.

| Method | Success Rate (%, ↑) / Speed up (↑) | | | | Average | FLOPs (%, ↓) |
| | Goal | Object | Spatial | Long | | |
|---|---|---|---|---|---|---|
| OpenVLA | 75.40 / 1.00 | 86.20 / 1.00 | 83.80 / 1.00 | 53.00 / 1.00 | 74.60 / 1.00 | 100 |
| SparseVLM | 74.20 / 1.33 | 84.00 / 1.33 | 83.40 / 1.33 | 52.80 / 1.33 | 73.60 / 1.33 | 75.55 |
| FoPru + R | 59.80 / 1.29 | 81.20 / 1.29 | 71.60 / 1.30 | 26.20 / 1.35 | 59.70 / 1.31 | 77.20 |
| PruMerge + R | 0.00 / 1.54 | 0.00 / 1.32 | 0.00 / 1.27 | 0.00 / 1.32 | 0.00 / 1.36 | 73.63 |
| FastVLM + R + S | 73.20 / 1.21 | 77.00 / 1.11 | 79.80 / 1.12 | 36.60 / 1.20 | 66.65 / 1.16 | 86.22 |
| VisionZip + R + S | 46.00 /1.20 | 47.40 / 1.23 | 34.20 / 1.19 | 4.60 / 1.22 | 33.05 / 1.21 | 81.95 |
| Ours (Speed) | 73.60 / **1.66** | 82.40 / **1.44** | 80.00 / **1.47** | 51.60 / **1.42** | 71.90 / **1.50** | **66.51** |
| Ours (Acc.) | **75.40** / 1.46 | **85.60** / 1.30 | **84.40** / 1.30 | **54.20** / 1.32 | **74.90** / 1.35 | 73.64 |

where $\mathbf{Q}, \mathbf{K}, \mathbf{V} \in \mathbb{R}^{N \times d_k}$, $\mathbf{W}_q, \mathbf{W}_k$, and $\mathbf{W}_v \in \mathbb{R}^{d \times d_k}$ are trainable weight matrices, and $N$ is the sequence length. The cumulative importance score of the tokens is given by:

$$\mathbf{Attn} = \mathrm{Softmax}\left(\frac{\mathbf{Q}\mathbf{K}^\top}{\sqrt{d_k}}\right)\mathbf{V}, \quad \mathbf{AccuAttn} = \frac{1}{N}\left(\mathbf{e}^\top \otimes \mathbf{I}_M\right)\mathrm{vec}(\mathbf{Attn}), \qquad (6)$$

where $\mathbf{Attn} \in \mathbb{R}^{N \times N}$ denotes the attention weight matrix, $\mathrm{vec}(\mathbf{Attn}) \in \mathbb{R}^{N^2 \times 1}$ is the column-wise vectorization, $\mathbf{e} \in \mathbb{R}^{N \times 1}$ is an all-one vector, and $\otimes$ denotes the Kronecker product. Based on this, we first identify semantically relevant tokens $\mathbf{T_{se}}$ by selecting those with cumulative attention scores exceeding a threshold $t_{k_s}$, i.e., $\mathbf{T_{se}} = \{\mathbf{x}_i \mid \mathbf{AccuAttn}_i > t_{k_s}\}$.

**Spatial-aware Token Importance.** We hypothesize that spatial information is primarily encoded in object contours. Therefore, we extract spatially informative tokens using the Canny edge detector. $\mathbf{X_s} = \mathrm{Canny}(\mathbf{X})$ denotes the edge-only image that preserves only contour information extracted from $\mathbf{X}$. We then obtain an ordered sequence of edge-based tokens using $\mathbf{T_{sp}} = f_E(\mathbf{X_s})$, where $f_E(\cdot)$ denotes the token extraction function.

Finally, the selected token set is obtained by computing the order-preserving union of the two, $\mathbf{T_{select}} = U(\mathbf{T_{se}}, \mathbf{T_{sp}})$, where $U(\cdot)$ denotes a union operation that preserves the original token ordering.

To align with the model collaboration strategy, we disable token pruning under low-speed conditions to avoid disrupting precise manipulations. Furthermore, motivated by the observation that higher motion speeds generally correspond to more *intuitive* actions, we define the pruning ratio to be positively correlated with the current velocity. Accordingly, the retained token ratio is defined as:

$$T_r(v) = \begin{cases} 1, & v < v_{p_{\min}} \\ 1 - \dfrac{v - v_{p_{\min}}}{v_{p_{max}} - v_{p_{\min}}}, & v \geq v_{p_{\min}} \end{cases}, \qquad (7)$$

where $v_{p_{\min}}$ is the minimum velocity threshold, $v_{p_{max}}$ is the maximum velocity of VLA model.

## 4 EXPERIMENTAL RESULTS

In this section, we present extensive experimental results to demonstrate the superior performance of SP-VLA. LIBERO consists of four task suites (Spatial, Object, Goal, and Long), covering 130 tasks with 2000 trajectories to evaluate model robustness. SimplerEnv provides three settings (Google-VM, Google-VA, and Bridge-VM) with variations in color, material, lighting, and camera pose for robustness assessment. We conduct real-world experiments by deploying SP-VLA on a Franka Research 3 robot to validate its practical effectiveness. In these experiments, we use CogACT as the base model and fine-tune it with 150 trajectories per task collected using the GELLO suite. Experiments are run on NVIDIA A100 GPUs (40GB), and all reported results are averaged over three independent runs with different random seeds to ensure statistical robustness.

Table 2: **The acceleration effects of individual modules.** As shown in the table, model scheduling yields the most significant acceleration for the VLA model with the least accuracy loss. In contrast, the model exhibits higher sensitivity to token pruning, and its performance collapses entirely when object edge information is removed.

| Method | Success Rate (%, ↑) / Speed up (↑) | | | | Average | FLOPs (%, ↓) |
|---|---|---|---|---|---|---|
| | Goal | Object | Spatial | Long | | |
| Ours | 75.40 / **1.46** | **85.60** / 1.30 | **84.40** / 1.30 | **54.20** / 1.32 | **74.90** / 1.35 | **73.64** |
| *w/o* Pruning | 74.40 / 1.23 | 84.20 / 1.27 | **84.00** / 1.18 | 53.30 / 1.39 | 73.98 / 1.27 | 78.75 |
| *w/o* Scheduling | **77.31** / 1.24 | 81.80 / 1.16 | 79.00 / 1.30 | 48.00 / 1.13 | 71.52 / 1.21 | 82.55 |
| *w/o* Canny | 33.60 / 1.34 | 39.00 / **1.35** | 22.00 / **1.35** | 1.10 / **1.37** | 23.93 / **1.35** | 73.40 |

Table 3: **Comparisons with the state-of-the-arts on SimplerEnv.** We apply SP-VLA to the LLM of CogACT, where the meanings of '+R' and '+S' follow Table 1. As shown, SP-VLAachieves SOTA performance across diverse tasks, delivering significant speedup while also improving accuracy.

| SIMPLER | Method | Success Rate (%, ↑) / Speed up (↑) | | | | Average | FLOPs (%, ↓) |
|---|---|---|---|---|---|---|---|
| | | PickCan | MoveNear | Drawer | DrawerApple | | |
| | CogACT | 91.30 / 1.00 | 85.00 / 1.00 | 71.80 / 1.00 | 50.90 / 1.00 | 74.80 / 1.00 | 100.00 |
| **Visual Matching** | Random Dropping | 9.70 / 1.20 | 20.40 / 1.20 | 53.50 / 1.20 | 0.00 / 1.20 | 20.90 / 1.20 | 58.50 |
| | FastV | 92.60 / 1.21 | 81.40 / 1.21 | 69.80 / 1.21 | 52.40 / 1.21 | 74.10 / 1.21 | 42.00 |
| | VLA-Cache | 92.00 / 1.38 | 83.30 / 1.38 | 70.50 / 1.38 | 51.60 / 1.38 | 74.40 / 1.38 | 80.10 |
| | EfficientVLA | **93.30** / 1.93 | 81.30 / 1.93 | 68.20 / 1.93 | **53.80** / 1.93 | 74.20 / 1.93 | **28.90** |
| | Ours | 90.00 / **2.62** | **82.08** / **2.52** | **75.35** / 1.80 | 52.78 / 1.67 | **75.05** / **2.15** | 38.15 |
| | CogACT | 89.60 / 1.00 | 80.80 / 1.00 | 28.30 / 1.00 | 46.60 / 1.00 | 61.30 / 1.00 | 100.00 |
| **Visual Aggregation** | Random Dropping | 4.00 / 1.20 | 16.10 / 1.20 | 15.60 / 1.20 | 0.00 / 1.20 | 8.90 / 1.20 | 58.50 |
| | FastV | 91.40 / 1.19 | 78.60 / 1.19 | 27.60 / 1.19 | 50.60 / 1.19 | 62.10 / 1.19 | 42.00 |
| | VLA-Cache | 91.70 / 1.37 | **79.30** / 1.37 | 32.50 / 1.37 | 45.80 / 1.37 | 62.30 / 1.37 | 82.60 |
| | EfficientVLA | **93.20** / 1.91 | 75.80 / 1.91 | 26.90 / **1.91** | **49.20** / 1.91 | 61.20 / 1.91 | **28.90** |
| | Ours | 86.18 / **2.48** | 77.33 / **2.63** | **55.29** / 1.81 | 41.80 / 1.44 | **65.16** / **2.09** | 40.20 |

| SIMPLER | Method | Success Rate (%, ↑) / Speed up (↑) | | | | Average | FLOPs (%, ↓) |
|---|---|---|---|---|---|---|---|
| | | PutSpoon | PutCarrot | StackBlock | PutEggplant | | |
| | CogACT | 71.70 /1.00 | 50.80 / 1.00 | 15.00 / 1.00 | 67.50 / 1.00 | 51.30 / 1.00 | 100.00 |
| **WindowX** | Random Dropping | 52.17 / 1.20 | 39.13 / 1.20 | 8.69 / 1.20 | 26.08 / 1.20 | 29.70 / 1.20 | 85.00 |
| | FoPru + R | 52.17 / 1.33 | 39.13 / 1.33 | 13.04 / 1.33 | 69.56 / 1.33 | 43.38 / 1.33 | 78.00 |
| | FastVLM + R + S | 34.78 / 1.14 | 30.43 / 1.14 | 4.35 / 1.14 | 30.43 / 1.08 | 25.00 / 1.13 | 90.50 |
| | Ours | **70.83** / **3.64** | **54.17** / 1.73 | **29.17** / **2.54** | **75.00** / 1.72 | **57.29** / **2.41** | **35.35** |

**Parameter Settings.** In this experiment, we set the buffer size to $n = 6$ and the deliberation ratio to $\tau = 0.5$. The choice of velocity thresholds is device dependent. In practice, one should first determine the maximum and minimum task-execution speeds of the embodied system, and then use 1/4 and 3/4 of this range as the values for $V_{\min}$ and $V_{\max}$, respectively. For example, in simulation, we assign a set of parameters for SP-VLA to a broad class of tasks, since different task types may require different configurations. In the real-world experiments of this paper, however, a single parameter set is sufficient for the Franka robot.

## 4.1 SIMULATION RESULTS

Table 1 and 3 report the main results of SP-VLA. SP-VLA achieves the best results in LIBERO, consistently delivering a 1.35× speedup without accuracy loss, and up to 1.5× faster inference with a slight 3% accuracy drop. In SimplerEnv tasks, it further achieves a 2× speedup with improved performance, indicating a degree of error-correction capability. Notably, on the Visual Aggregation Drawer task, SP-VLA improves performance by about 27% while achieving a 1.8× speedup, demonstrating clear error-correction capability. Besides, the VLA model is highly sensitive to spatial information: disrupting token order or retaining too few tokens severely impairs perception, often leading to task failure, particularly with generic compression methods. To address this, we reorder tokens and add position tokens ('+R' and '+S'), but these approaches still perform poorly with substantial performance degradation. Even after supplementing positional information and preserving token order, FoPru (Jiang et al., 2024) and VisionZip (Yang et al., 2024) still suffer from substantial

Table 4: **Real-robot performance on Franka.** We evaluate two manipulation tasks over 50 trials each (20 morning, 10 noon, 20 evening). SP-VLA achieves over **2.5×** acceleration on average while preserving accuracy, reduces FLOPs by over **60%**, and lowers inference latency by nearly **50%**.

| Method | Success Rate (%, ↑) / Speed Up (↑) | | Average | FLOPs (%, ↓) | Latency (s, ↓) |
| | Pick Up | Pick and Place | | | |
|---|---|---|---|---|---|
| CogACT | 80.00 / 1.00 | 74.00 / 1.00 | 77.00 / 1.00 | 100 | 0.27 |
| SP-VLA | 78.00 / **2.46** | 74.00 / **2.57** | 76.00 / **2.52** | **35.55** | **0.13** |

Table 5: **Latency and Frequency in the SimplerEnv Environment.** We measure these metrics on an RTX 4090. SP-VLA achieves approximately a 2.2× inference speedup, highlighting its capability for phase-aware dynamic acceleration that adapts frequency to different stages of the task.

| SIMPLER | Evaluation Indicator | Frequency (Hz, ↑) / Latency (s, ↓) | | | | Average |
| | | PickCan | MoveNear | Drawer | DrawerApple | |
|---|---|---|---|---|---|---|
| Visual Matching | CogACT | 4.00 / 0.25 | 4.00 / 0.25 | 3.85 / 0.26 | 3.23 / 0.31 | 3.77 / 0.27 |
| | SP-VLA | 9.09 / 0.11 | 8.33 / 0.12 | 7.14 / 0.14 | 7.69 / 0.13 | 8.06 / 0.13 |

| SIMPLER | Evaluation Indicator | Frequency (Hz, ↑) / Latency (s, ↓) | | | | Average |
| | | PutSpoon | PutCarrot | StackBlock | PutEggplant | |
|---|---|---|---|---|---|---|
| WindowX | CogACT | 4.00 / 0.25 | 4.00 / 0.25 | 4.00 / 0.25 | 3.85 / 0.26 | 3.96 / 0.25 |
| | SP-VLA | 12.5 / 0.08 | 6.25 / 0.16 | 8.33 / 0.12 | 7.69 / 0.13 | 8.69 / 0.12 |

performance degradation with limited speedup. These results demonstrate that SP-VLA effectively identifies both temporal and spatial redundancies in VLA models, and accelerates the inference process through joint model scheduling and token pruning, while preserving the model's spatial perception capabilities and overall performance. This confirms the significant effectiveness of our approach in accelerating VLA inference.

## 4.2 REAL-WORLD RESULTS

To further validate the acceleration gains of SP-VLA, we first train CogACT using 150 trajectories collected for each of the two tasks—*Pick up the cylinder* and *Pick and Place the cylinder*. We then deploy SP-VLA on a real Franka Research 3 robot to evaluate its performance. The hyperparameters are set using 1/4 and 3/4 of Franka's velocity range as $V_{min}$ and $V_{max}$, respectively, with $\tau = 0.5$ and $n = 6$. For each task, we conduct 50 evaluations (20 in the morning, 10 at noon, and 20 in the evening) and report the average performance to approximate operation across different time periods. As shown in Table 4, CogACT achieves success rates of 80% and 74% on the two tasks. After integrating SP-VLA, the success rates become 78% and 74%. This reflects only a 1% decrease in average accuracy while achieving a 2.52× acceleration. These results demonstrate that SP-VLA provides stable and consistent acceleration in both simulation and real-world deployment.

## 4.3 ANALYSIS

**The Acceleration Effects of Individual Modules.** The ablation results are summarized in Table 2. Specifically, we conduct three ablation experiments here: (1) disabling token pruning, (2) removing model scheduling, and (3) excluding positional information from the token pruning process. Overall, model scheduling emerges as the most effective component for accelerating the VLA model, achieving a 1.27× speedup with only a 1% drop in accuracy. This indicates that VLA models contain substantial temporal redundancy. In contrast, token pruning yields only moderate acceleration but results in a substantial accuracy drop, indicating that the VLA model is highly sensitive to token reduction and offers limited redundancy for compression. Notably, eliminating the Canny edge information results in a dramatic 50% degradation in performance, effectively collapsing the model's functionality. This underscores the critical role of relative token positions and object contour information in enabling the VLA model's spatial perception. Ultimately, the joint application of these techniques yields the best overall acceleration, with no compromise in model accuracy.

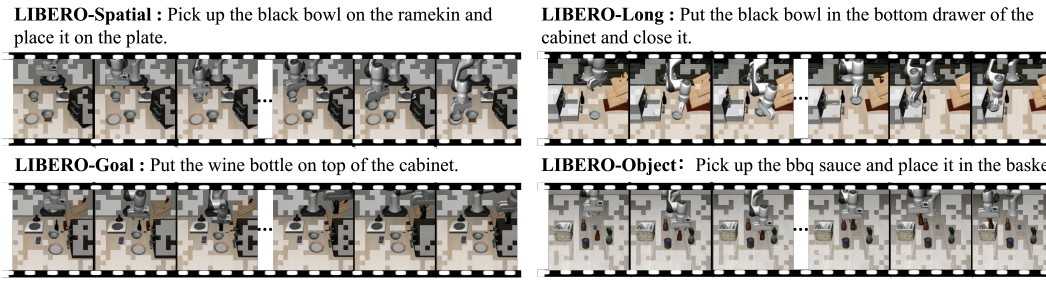

Figure 4: **Visualizations of SP-VLA across different tasks.** As shown in the figure, our method efficiently prunes redundant image regions to accelerate VLA inference while preserving key object contours to maintain spatial perception.

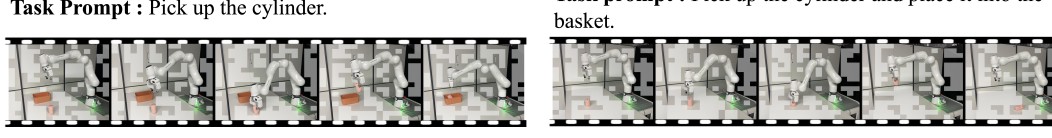

Figure 5: **Visualizations of SP-VLA on real tasks.** As illustrated in the figure, our method selectively prunes unnecessary tokens while retaining critical visual cues, enabling efficient acceleration in real-world scenarios without sacrificing task-relevant information.

**Frequency and Latency.** We evaluate the inference frequency and latency of SP-VLA on a single NVIDIA RTX 4090 (40GB), with results in Table 5. CogACT runs at less than 4 Hz, while our method increases the frequency to over 8 Hz—a 2.2× improvement—along with a 2.2× reduction in per-inference latency. Furthermore, on LIBERO, our method delivers a 1.4× frequency improvement, indicating that VLA models exhibit substantial temporal and spatial redundancy, which our approach effectively exploits for dynamic frequency acceleration. More detailed results are in Appendix A.2.

**Visualizations.** To evaluate the efficiency of SP-VLA, we present both simulation and real-world visualizations in Fig. 4 and Fig. 5, illustrating how the method accelerates VLA inference while preserving task-relevant information. Fig. 4 shows how SP-VLA performs task completion across various tasks during the invocation of the model. As shown in the figure, SP-VLA adaptively identifies redundant components and prunes tokens accordingly to accelerate inference. Morever, it effectively preserves object edge information, maintaining the spatial perception capability required by the model. These results demonstrate the effectiveness of our spatio-semantic dual-aware token pruning method in significantly enhancing inference efficiency. Fig. 5 illustrates the real-world performance of SP-VLA. As shown in the figure, our method effectively identifies and removes redundant tokens while preserving the essential content, enabling efficient acceleration without compromising task-relevant information. This demonstrates that SP-VLA preserves the execution accuracy of the base model while simultaneously achieving substantial acceleration. For details on parameter sensitivity analysis, in-depth experimental analysis, the acceleration ratios of different model parts and additional experiments and visualizations, please refer to Appendices A.3, A.4, A.5, A.7, A.8 and A.9.

## 5 CONCLUSION

In this work, we propose SP-VLA, a unified framework that accelerates VLA models through joint model scheduling and token pruning. By dynamically switching between a full VLA model and a lightweight generator based on *deliberative* or *intuitive* actions, and pruning tokens according to spatio-semantic importance during VLA invocation, SP-VLA enables frequency-aware and task-adaptive acceleration. Extensive experiments demonstrate that our method achieves 1.5× lossless speedup in LIBERO and 2.4× in SimplerEnv, with up to 6% performance gain, while simultaneously improving both inference frequency and latency by 1.4× in LIBERO and 2.2× in SimplerEnv.

## 6 ETHICS STATEMENT

This work focuses solely on accelerating embodied models and involves only robotic arm simulations and real-world experiments. It does not involve human subjects, sensitive personal data, or ethically concerning applications. The research does not present foreseeable risks of misuse or harm, nor does it raise issues of bias, discrimination, privacy, or security. All experiments were conducted in compliance with standard research integrity practices.

## 7 REPRODUCIBILITY STATEMENT

We have taken several measures to ensure the reproducibility of our results. All experiments were conducted with multiple random seeds and repeated runs to confirm robustness. The backbone models and checkpoints used in this work are publicly available and open-source. The experimental environments (LIBERO and SimplerEnv) are standardized and publicly accessible. All hyperparameters and implementation details will be fully documented and released in a public GitHub repository, and additional sensitivity analyses and extended results are provided in the appendix to further support reproducibility.

## 8 ACKNOWLEDGMENTS

This work was supported by National Natural Science Foundation of China (Grant No. 92467204, 62472249 and 62402264), and Shenzhen Science and Technology Program (Grant No. JCYJ20220818101014030 and KJZD20240903102300001).

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

# A APPENDIX

## A.1 THE USE OF LARGE LANGUAGE MODELS (LLMS)

In this work, Large Language Models (LLMs) were used solely to polish the language for clarity and readability. No LLMs were employed for idea generation, experimental design, data analysis, or any other part of the research process.

## A.2 LATENCY AND FREQUENCY

Table 6: **Latency and Frequency in the LIBERO Environment.** We measure these metrics on an RTX 4090. As shown in the table, SP-VLA achieves about a 1.4× inference speedup.

| Evaluation Inidcator | Frequency (Hz, ↑) / latency (s, ↓) | | | | Average |
|---|---|---|---|---|---|
| | **Goal** | **Object** | **Spatial** | **Long** | |
| **OpenVLA** | 3.13 / 0.32 | 3.27 / 0.31 | 2.99 / 0.33 | 3.32 / 0.30 | 3.18 / 0.32 |
| **SP-VLA** | 4.34 / 0.23 | 4.56 / 0.22 | 4.37 / 0.23 | 4.44 / 0.23 | 4.43 / 0.23 |

Table 7: **Latency and Frequency in the SimplerEnv Environment.** We measure these metrics on an RTX 4090. As shown in the table, SP-VLA achieves about a 2.2× inference speedup.

| SIMPLER | Evaluation Indicator | Frequency (Hz, ↑) / latency (s, ↓) | | | | Average |
|---|---|---|---|---|---|---|
| | | **PickCan** | **MoveNear** | **Drawer** | **DrawerApple** | |
| **Visual Matching** | CogACT | 4.00 / 0.25 | 4.00 / 0.25 | 3.85 / 0.26 | 3.23 / 0.31 | 3.77 / 0.27 |
| | SP-VLA | 9.09 / 0.11 | 8.33 / 0.12 | 7.14 / 0.14 | 7.69 / 0.13 | 8.06 / 0.13 |
| **Visual Aggregation** | CogACT | 4.17 / 0.24 | 3.85 / 0.26 | 3.45 / 0.29 | 2.86 / 0.35 | 3.58 / 0.29 |
| | SP-VLA | 9.09 / 0.11 | 9.09 / 0.11 | 5.56 / 0.18 | 4.76 / 0.21 | 7.13 / 0.15 |
| SIMPLER | Evaluation Indicator | Frequency (Hz, ↑) / latency (s, ↓) | | | | Average |
| | | **PutSpoon** | **PutCarrot** | **StackBlock** | **PutEggplant** | |
| **WindowX** | CogACT | 4.00 / 0.25 | 4.00 / 0.25 | 4.00 / 0.25 | 3.85 / 0.26 | 3.96 / 0.25 |
| | SP-VLA | 12.5 / 0.08 | 6.25 / 0.16 | 8.33 / 0.12 | 7.69 / 0.13 | 8.69 / 0.12 |

Tables 6 and 7 report the frequency and latency measurements of SP-VLA on an RTX 4090. As shown in the table, OpenVLA and CogACT achieve only 3–4 Hz inference frequency. By incorporating SP-VLA, their inference rates are improved by approximately 1.4× and 2.2×, respectively, while preserving model accuracy. This suggests that VLA models contain substantial temporal and spatial redundancies, offering considerable acceleration potential. By effectively identifying and exploiting both, SP-VLA achieves significant speedups and demonstrates strong application prospects.

## A.3 SENSITIVITY ANALYSIS OF KEY PARAMETERS.

Table 8 presents an ablation study in the LIBERO simulation environment, examining the impact of execution speed $V$, buffer size $n$, and the proportion of deliberative actions $\tau$. Specifically, execution speed is varied by $\pm 25\%$ around $V_{max}$ and $V_{min}$, buffer size is set to $\{4, 6, 8\}$, and the proportion of deliberative actions in the buffer is chosen from $\{3/8, 4/8, 5/8\}$. The results show how model accuracy and speedup vary under different settings. Overall, SP-VLA is robust to speed changes, with $\pm 25\%$ fluctuations having little impact on accuracy or acceleration. In contrast, buffer size $n$ and the proportion of intuitive actions $\tau$ have a much stronger effect, significantly influencing accuracy. This is because, even during skipping, SP-VLA relies on the VLA model to provide correct directional guidance and critical decision points, thereby preserving overall performance.

## A.4 ANALYSIS OF EXPERIMENTAL RESULTS.

**Analysis for Baseline Methods.** Owing to the lack of a direct baseline, we instead compare our method with state-of-the-art approaches that have demonstrated strong performance in LLAVA. Tables 1 and 3 show that the majority of baseline methods do not perform well on VLA models.

Table 8: **Sensitivity Analysis of Key Parameters.** We present the sensitivity analysis of SP-VLA in the LIBERO environment. We vary execution speed ($\pm 25\%$ around $V_{\max}$ and $V_{\min}$), buffer size ($n \in 4, 6, 8$), and the proportion of deliberative actions ($\tau \in 3/8, 4/8, 5/8$), and report their effects on model accuracy and acceleration. The results show that SP-VLA is robust to speed changes, whereas buffer size and the proportion of deliberative actions exert a stronger influence, significantly affecting accuracy.

| Task | Scaling | Success Rate (%, ↑) / Speed up (↑) | | | |
| --- | --- | --- | --- | --- | --- |
| | | $V_{min}$ **(0.2)** | $V_{max}$ **(0.5)** | $\tau$ **(0.5)** | **n (6)** |
| **Object** | 0 | 82.4 / **1.44** | 82.4 / 1.44 | **82.4** / 1.44 | **82.4** / 1.44 |
| | ↑ [25%, 25%, 5/8, 8] | 82.2 / 1.41 | **83.2** / 1.33 | 81.7 / 1.34 | 68.4 / **1.58** |
| | ↓ [25%, 25%, 3/8, 4] | **83.6** / 1.37 | 81.6 / **1.38** | 64.4 / **1.75** | 80.8 / 1.33 |
| **Goal** | 0 | 73.6 / **1.66** | 73.6 / 1.66 | 73.6 / 1.66 | 73.6 / 1.66 |
| | ↑ [25%, 25%, 5/8, 8] | 72.8 / 1.41 | 70.4 / 1.47 | 71.7 / 1.45 | 66.2 / **1.87** |
| | ↓ [25%, 25%, 3/8, 4] | **74.8** / 1.22 | 69.0 / **1.78** | 56.4 / **1.91** | **74.0** / 1.31 |
| **Spatial** | 0 | **80.0** / 1.47 | **80.0** / 1.47 | **80.0** / 1.47 | **80.0** / 1.47 |
| | ↑ [25%, 25%, 5/8, 8] | 77.6 / **1.49** | **80.0** / 1.47 | 79.1 / 1.36 | 74.8 / **1.59** |
| | ↓ [25%, 25%, 3/8, 4] | 78.6 / 1.43 | 78.4 / **1.52** | 62.2 / **1.83** | 74.4 / 1.27 |
| **Long** | 0 | **51.6** / 1.42 | **51.6** / 1.42 | **51.6** / 1.42 | **51.6** / 1.42 |
| | ↑ [25%, 25%, 5/8, 8] | 50.4 / 1.45 | 49.2 / 1.31 | 48.9 / 1.28 | 47.4 / **1.37** |
| | ↓ [25%, 25%, 3/8, 4] | 50.1 / **1.37** | 46.6 / **1.53** | 42.5 / **1.77** | 44.6 / 1.32 |

Although the baseline methods are effective lightweight solutions for VLMs, VLA models differ in two fundamental aspects: they are highly sensitive to spatial information and are required to perform sequential decision-making. Consequently, VLA models exhibit both temporal and spatial redundancies. These characteristics render token lightweighting strategies designed for VLMs unfit for VLA applications.

(1) **Failed extraction of spatial information.** Since FoPru (Jiang et al., 2024) and PruMerge (Shang et al., 2024) already incorporate mechanisms for extracting spatial information, we did not add Canny-based edge detection to these methods. However, their spatial information extraction primarily aims to preserve semantic integrity rather than focusing on the relative positioning between objects, which leads to performance degradation when applied to VLA models.

(2) **Feature fusion–induced spatial ambiguity.** VisionZip (Yang et al., 2024) merges tokens to compress semantic information, which proves effective for VLMs but disrupts spatial information in VLA models, leading to a collapse in performance.

(3) **Performance degradation due to overlooked temporal redundancy.** Although FastVLM (Vasu et al., 2024) and FastV (Chen et al., 2024) performs token pruning and information augmentation, it retains a fixed number of tokens at each timestep, making it unable to adapt to the varying complexity of VLA tasks. This limitation results in degraded performance. In contrast, SP-VLA maintains a constant cumulative attention threshold, dynamically adjusting the number of retained tokens—preserving more at critical moments and fewer during less demanding phases—thereby effectively preserving the performance of the VLA model.

In contrast, VLA-Cache (Xu et al., 2025) and EfficientVLA (Yang et al., 2025) are acceleration algorithms specifically designed for VLA models. VLA-Cache leverages KV cache reuse, while EfficientVLA combines layer skipping, token pruning, and activation caching to accelerate both the VLM and the Action Head, achieving relatively stronger speedups. SP-VLA, on the other hand, targets temporal and spatial redundancies without altering the model itself and is therefore, to some extent, orthogonal to these two methods.

**Analysis for the Simulation Environments.** We evaluate SP-VLA using two simulation environments: LIBERO (Liu et al., 2023) and SimplerEnv (Li et al., 2024a). LIBERO consists of four task suites (Spatial, Object, Goal, and Long), covering 130 tasks with 2000 trajectories to evaluate model robustness. SimplerEnv provides three settings (Google-VM, Google-VA, and Bridge-VM) with variations in color, material, lighting, and camera pose for robustness assessment. LIBERO is

Table 9: **The Acceleration Ratios of Model Scheduling and Token Pruning.** Model scheduling yields the most prominent speedup, with redundancy increasing as task sequences grow longer. Optimizing a single dimension reveals redundancy specific to it, suggesting that temporal and spatial accelerations are interdependent. SP-VLA exploits this interplay to achieve superior acceleration.

| Method | Pruning Rate (%, ↑) / Intuitive Action Rate (%, ↑) | | | | Average | Average Acc. (%, ↑) |
|---|---|---|---|---|---|---|
| | Goal | Object | Spatial | Long | | |
| Ours | 19.45 / 15.00 | 6.00 / 18.07 | 10.50 / 13.90 | 5.80 / 19.64 | 10.44 / 16.65 | **74.90** |
| *w/o* Pruning | 0.00 / **18.58** | 0.00 / **21.11** | 0.00 / **15.41** | 0.00 / **28.00** | 0.00 / **20.78** | 73.98 |
| *w/o* Scheduling | **19.66** / 0.00 | **13.50** / 0.00 | **22.70** / 0.00 | 11.53 / 0.00 | **16.85** / 0.00 | 71.52 |
| *w/o* Canny | 16.42 / 11.42 | 10.10 / 18.13 | 17.40 / 11.13 | **17.30** / 13.20 | 15.31 / 13.47 | 23.93 |

Table 10: **The results under different settings.** SP-VLA maintains high accuracy across various acceleration ratios, demonstrating the strong robustness of our approach. For example, our method not only improves accuracy under moderate acceleration (1.35×), but also maintains competitive performance under higher speedup (1.5×) with only a 3% accuracy drop.

| Method | Success Rate (%, ↑) / Speed up (↑) | | | | Average | FLOPs (%, ↓) |
|---|---|---|---|---|---|---|
| | Goal | Object | Spatial | Long | | |
| OpenVLA | 75.40 | 86.20 | 83.80 | 53.30 | 74.68 | 100.00 |
| Ours-1 | 73.60 / **1.66** | 82.40 / **1.44** | 80.00 / **1.47** | 51.60 / 1.42 | 71.90 / **1.50** | 66.51 |
| Ours-2 | 72.20 / 1.58 | 83.60 / 1.36 | 81.40 / 1.35 | 50.40 / **1.44** | 71.90 / 1.43 | 69.83 |
| Ours-3 | 74.80 / 1.36 | 84.80 / 1.30 | 82.20 / 1.29 | 53.40 / 1.29 | 73.80 / 1.31 | 76.25 |
| Ours-4 | **75.40** / 1.46 | **85.60** / 1.30 | **84.40** / 1.30 | **54.20** / 1.32 | **74.90** / 1.35 | 73.63 |

mainly used to evaluate the stability of SP-VLA across diverse tasks, whereas SimplerEnv focuses on generalization. As shown in Tables 1 and 3, SP-VLA delivers strong results in both settings: 1.5× lossless acceleration on LIBERO and up to 2.4× acceleration with a 6% performance gain on SimplerEnv, underscoring its robustness.

### A.5 THE ACCELERATION RATIOS OF MODEL SCHEDULING AND TOKEN PRUNING.

Using the same parameter settings as in Table 2, Table 9 presents the acceleration contributions of model scheduling and token pruning. Model scheduling achieves the highest accuracy at comparable acceleration levels, suggesting that much of the computational redundancy in VLA models arises from *intuitive* and *deliberate* actions. We further find that the proportion of *intuitive* actions grows with task length, from 18% in LIBERO-Spatial (1.18× speedup) to 28% in LIBERO-Long (1.39× acceleration). While individual techniques expose redundancy mainly along a single dimension—e.g., token pruning identifies 22.7% in LIBERO-Spatial and model scheduling 21% in LIBERO-Object—SP-VLA jointly optimizes temporal and spatial redundancies. These dimensions are interdependent rather than orthogonal, and SP-VLA effectively exploits this synergy to achieve superior acceleration.

### A.6 LIMITATION

In this work, we discovered and verified that the VLA model can be categorized into *deliberative* actions and *intuitive* actions. By leveraging this distinction, we propose a method that jointly schedules the model and prunes tokens for VLA models, achieving acceleration in both the temporal and spatial dimensions. However, our current exploration of *intuitive* action generation is limited to model lightweighting and remains preliminary. As a result, a complete separation between the generation of *deliberative* and *intuitive* actions has not yet been achieved. We believe that explicitly distinguishing these two types of actions in the behavioral logic of VLA models will make them more human-like and is a crucial step toward achieving higher accuracy, faster inference, and lower energy consumption, with significant potential for future development. Therefore, this will be one of the key directions of our future exploration.

Table 11: **The Acceleration Ratios of Model Scheduling and Token Pruning.** Overall, model scheduling achieves higher acceleration ratios than token pruning, indicating that VLA contains significant temporal redundancy. By categorizing actions into *deliberative* and *intuitive* types, this redundancy can be efficiently addressed. Moreover, the varying tolerance of different tasks to the two acceleration methods suggests that these approaches are not orthogonal, but rather complementary, working together to achieve optimal performance.

| Method | Pruning Rate (%, ↑) / Intuitive Action Rate (%, ↑) | | | | Average | Average Acc. (↑) |
| | Goal | Object | Spatial | Long | | |
|---|---|---|---|---|---|---|
| Ours-1 | 19.45 / 15.00 | 6.00 / 18.07 | 10.50 / 13.90 | 5.80 / 19.64 | 10.44 / 16.65 | 71.90 |
| Ours-2 | 24.91 / 15.88 | 6.25 / 21.58 | 15.51 / 12.63 | **9.09** / 23.81 | 13.93 / 18.48 | 72.20 |
| Ours-3 | 15.01 / 12.87 | 5.89 / 18.61 | 9.57 / 14.33 | 5.71 / 17.95 | 9.05 / 15.94 | 73.80 |
| Ours-4 | **26.70 / 17.40** | **6.47 / 25.70** | **18.02 / 17.29** | 5.67 / **25.53** | **14.22 / 21.48** | 74.90 |

## A.7 MORE EXPERIMENTAL RESULTS.

Tables 10 and 11 present the results of SP-VLA under different settings. Overall, SP-VLA maintains high accuracy across diverse acceleration ratios, demonstrating robust adaptability to varying acceleration demands. For instance, at the $1.35\times$ acceleration rate, SP-VLA achieves an overall accuracy of 74.90%, outperforming OpenVLA. This indicates that moderately reducing redundancy can help correct errors and improve performance. Even under a $1.5\times$ acceleration, the accuracy only drop 3%, highlighting the significant temporal and spatial redundancy present in VLA models. These results highlight that SP-VLA achieves outstanding acceleration on VLA models, while also demonstrating strong robustness by maintaining stable performance across a wide range of acceleration ratios.

In terms of acceleration contributions, model scheduling delivers markedly larger speedups than token pruning, consistently outperforming it across all settings. This finding suggests that VLA execution involves a high proportion of *intuitive* actions, revealing substantial computational redundancy. On the other hand, different tasks exhibit varying levels of tolerance to acceleration methods. For example, the LIBERO-Goal task allows a relatively large degree of token pruning, achieving a 26.70% reduction while maintaining accuracy. In contrast, the LIBERO-Long task only supports a 5.67% pruning rate, with most of the acceleration coming from *intuitive* action generation. This indicates that as task complexity increases, higher spatial perception capability is required from the VLA model. Furthermore, the two acceleration methods are not entirely independent; instead, they work synergistically to achieve optimal acceleration performance.

To further demonstrate the generality and effectiveness of our approach, we provide visualizations and task-wise speedup curves for various LIBERO tasks in Sections A.8 and A.9, respectively. These examples further illustrate that SP-VLA is well-suited for a wide range of manipulation tasks and achieves remarkable acceleration performance.

## A.8 MORE VISUALIZATION RESULTS.

**LIBERO-Spatial :** Pick up the black bowl between the plate and the ramekin and place it on the plate.

**LIBERO-Spatial :** Pick up the black bowl next to the ramekin and place it on the plate.

**LIBERO-Spatial :** Pick up the black bowl from table center and place it on the plate.

**LIBERO-Spatial :** Pick up the black bowl on the cookie box and place it on the plate.

**LIBERO-Spatial :** Pick up the black bowl in the top drawer of the wooden cabinet and place it on the plate.

**LIBERO-Spatial :** Pick up the black bowl on the ramekin and place it on the plate.

**LIBERO-Spatial :** Pick up the black bowl next to the cookie box and place it on the plate.

**LIBERO-Spatial :** Pick up the black bowl on the stove and place it on the plate.

**LIBERO-Spatial :** Pick up the black bowl next to the plate and place it on the plate.

**LIBERO-Spatial :** Pick up the black bowl on the wooden cabinet and place it on the plate.

Figure 6: **Visualization examples generated by SP-VLA on LIBERO-Spatial.**

**LIBERO-Goal:** Open the middle drawer of the cabinet.

**LIBERO-Goal:** Put the bowl on the stove.

**LIBERO-Goal:** Put the wine bottle on top of the cabinet.

**LIBERO-Goal:** Open the top drawer and put the bowl inside.

**LIBERO-Goal:** Put the bowl on top of the cabinet.

**LIBERO-Goal:** Push the plate to the front of the stove.

**LIBERO-Goal:** Put the cream cheese in the bowl.

**LIBERO-Goal:** Turn on the stove.

**LIBERO-Goal:** Put the bowl on the plate.

**LIBERO-Goal:** Put the wine bottle on the rack.

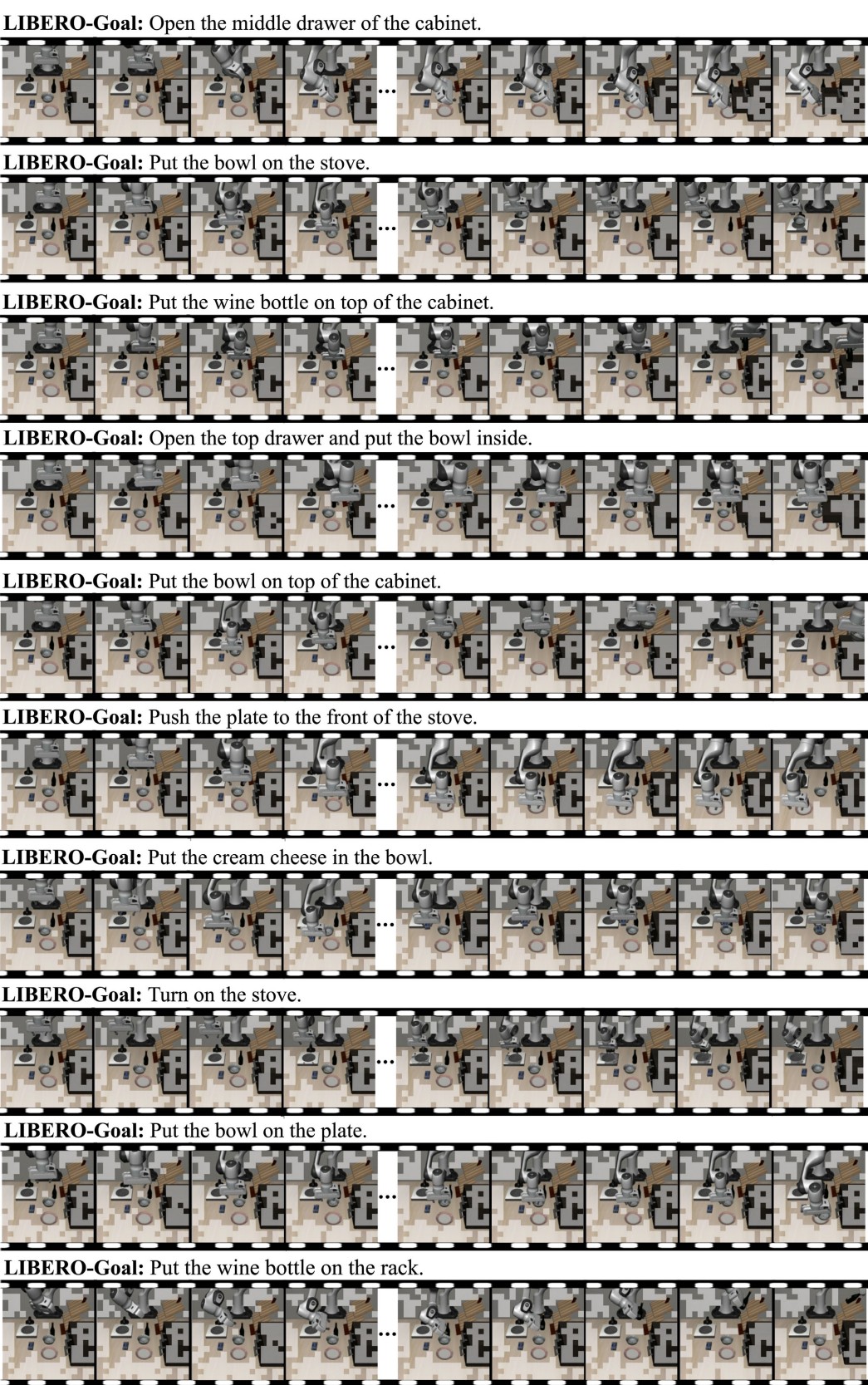

Figure 7: **Visualization examples generated by SP-VLA on LIBERO-Goal.**

**LIBERO-Object:** Pick up the alphabet soup and place it in the basket.

**LIBERO-Object:** Pick up the cream cheese and place it in the basket.

**LIBERO-Object:** Pick up the salad dressing and place it in the basket.

**LIBERO-Object:** Pick up the bbq sauce and place it in the basket.

**LIBERO-Object:** Pick up the ketchup and place it in the basket.

**LIBERO-Object:** Pick up the tomato sauce and place it in the basket.

**LIBERO-Object:** Pick up the butter and place it in the basket.

**LIBERO-Object:** Pick up the milk and place it in the basket.

**LIBERO-Object:** Pick up the chocolate pudding and place it in the basket.

**LIBERO-Object:** Pick up the orange juice and place it in the basket.

Figure 8: **Visualization examples generated by SP-VLA on LIBERO-Object.**

**LIBERO-Long:** Put both the alphabet soup and the tomato sauce in the basket.

**LIBERO-Long :** Put both the cream cheese box and the butter in the basket.

**LIBERO-Long:** Turn on the stove and put the moka pot on it.

**LIBERO-Long :** Put the black bowl in the bottom drawer of the cabinet and close it.

**LIBERO-Long:** Put the white mug on the left plate and put the yellow and white mug on the right plate.

**LIBERO-Long:** Pick up the book and place it in the back compartment of the caddy.

**LIBERO-Long:** Put the white mug on the plate and put the chocolate pudding to the right of the plate.

**LIBERO-Long:** Put both the alphabet soup and the cream cheese box in the basket.

**LIBERO-Long:** Put both moka pots on the stove.

**LIBERO-Long:** Put the yellow and white mug in the microwave and close it.

Figure 9: **Visualization examples generated by SP-VLA on LIBERO-Long.**

## A.9 Action Dynamics across Different Tasks.

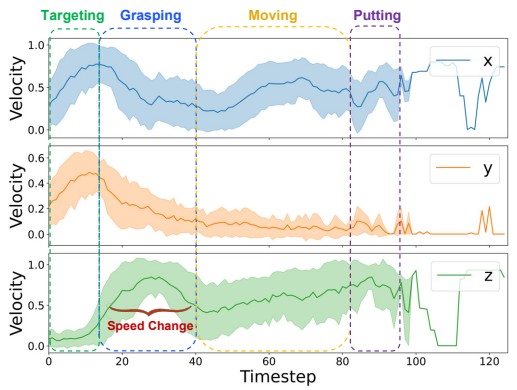

(a) **LIBERO-Spatial**: Pick up the black bowl between the plate and the ramekin and place it on the plate.

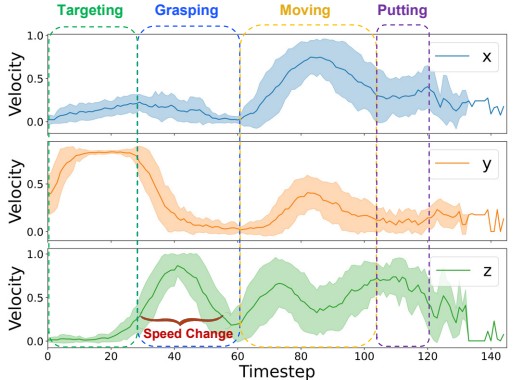

(b) **LIBERO-Spatial**: Pick up the black bowl next to the ramekin and place it on the plate.

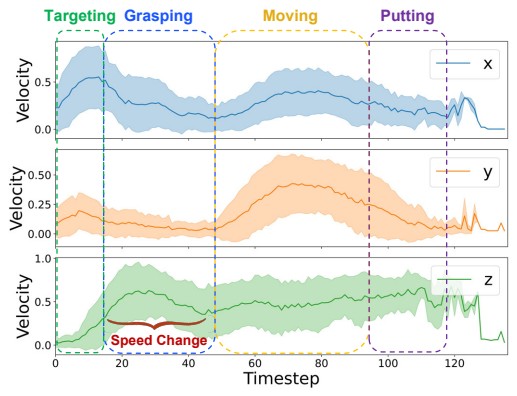

(c) **LIBERO-Spatial**: Pick up the black bowl from table center and place it on the plate.

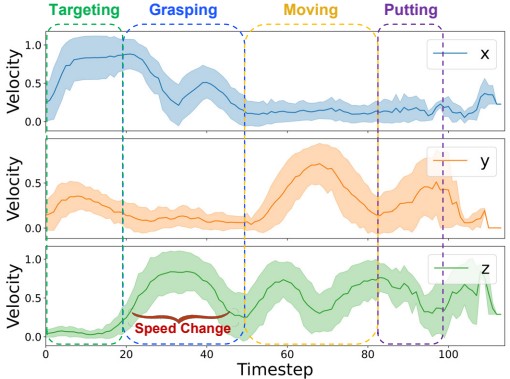

(d) **LIBERO-Spatial**: Pick up the black bowl on the cookie box and place it on the plate.

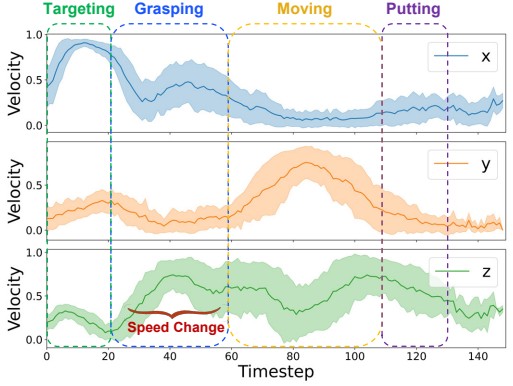

(e) **LIBERO-Spatial**: Pick up the black bowl in the top drawer of the wooden cabinet and place it on the plate.

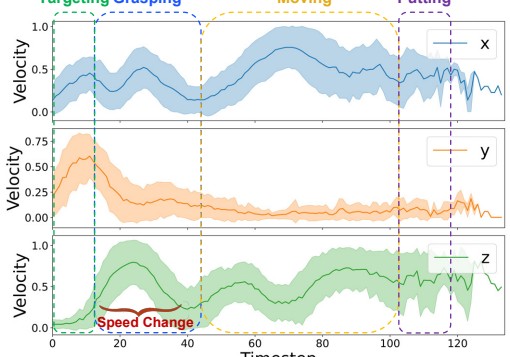

(f) **LIBERO-Spatial**: Pick up the black bowl on the ramekin and place it on the plate.

Figure 10: **Visualizations of SP-VLA on the first 6 LIBERO-Spatial tasks.**

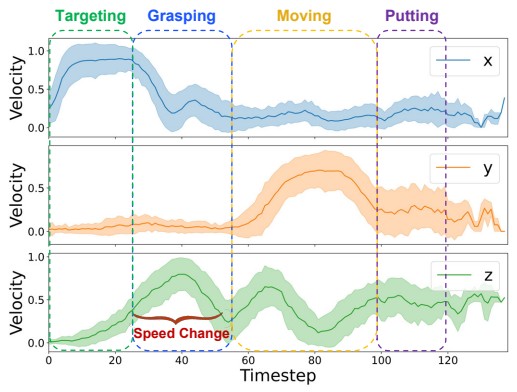

(a) **LIBERO-Spatial**: Pick up the black bowl next to the cookie box and place it on the plate.

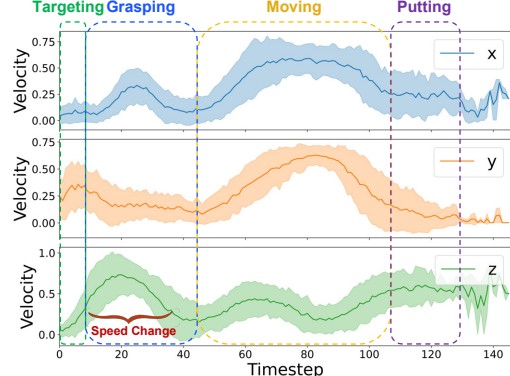

(b) **LIBERO-Spatial**: Pick up the black bowl on the stove and place it on the plate.

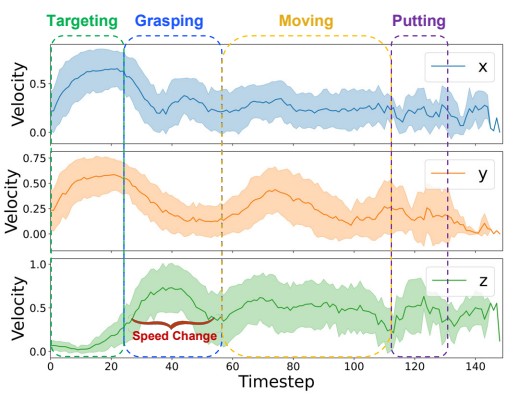

(c) **LIBERO-Spatial**: Pick up the black bowl next to the plate and place it on the plate.

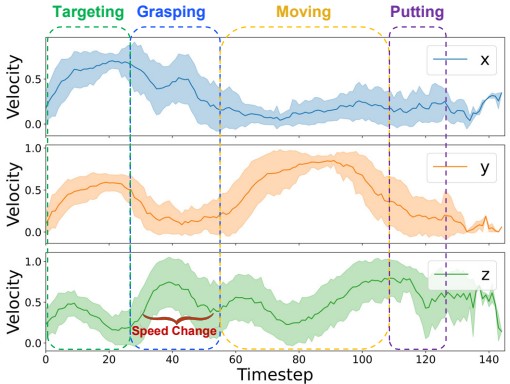

(d) **LIBERO-Spatial**: Pick up the black bowl on the wooden cabinet and place it on the plate.

Figure 11: **Visualizations of SP-VLA on the remaining 4 LIBERO-Spatial tasks.**

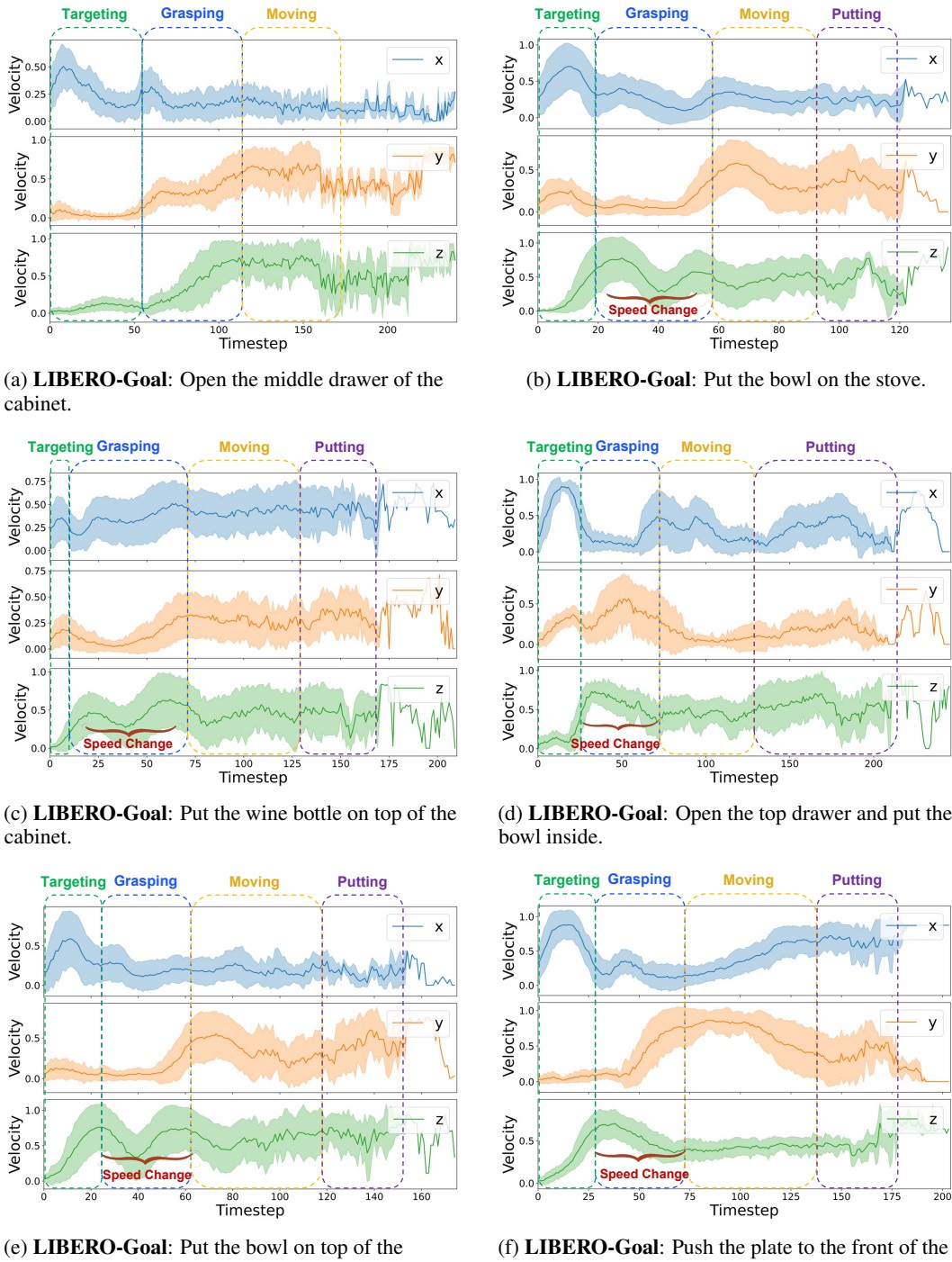

(a) **LIBERO-Goal**: Open the middle drawer of the cabinet.

(b) **LIBERO-Goal**: Put the bowl on the stove.

(c) **LIBERO-Goal**: Put the wine bottle on top of the cabinet.

(d) **LIBERO-Goal**: Open the top drawer and put the bowl inside.

(e) **LIBERO-Goal**: Put the bowl on top of the cabinet.

(f) **LIBERO-Goal**: Push the plate to the front of the stove.

Figure 12: **Visualizations of SP-VLA on the first 6 LIBERO-Goal tasks.**

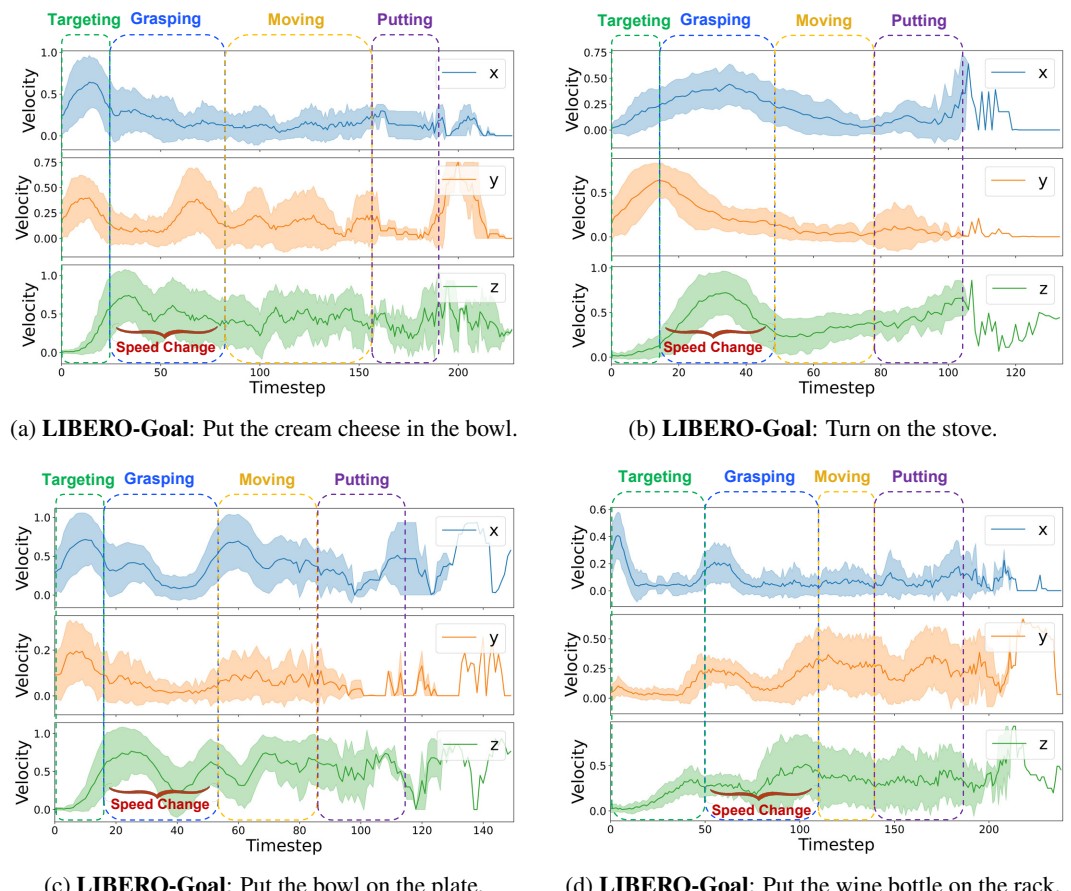

(a) **LIBERO-Goal**: Put the cream cheese in the bowl.

(b) **LIBERO-Goal**: Turn on the stove.

(c) **LIBERO-Goal**: Put the bowl on the plate.

(d) **LIBERO-Goal**: Put the wine bottle on the rack.

Figure 13: **Visualizations of SP-VLA on the remaining 4 LIBERO-Goal tasks.**

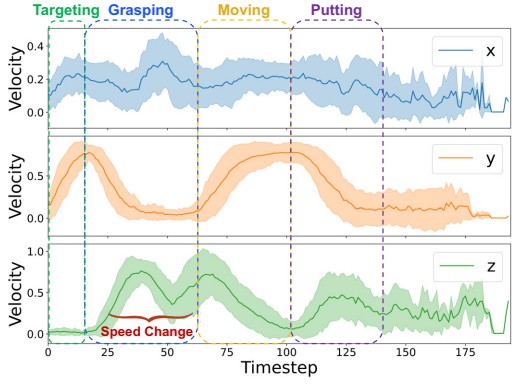

(a) **LIBERO-Object**: Pick up the alphabet soup and place it in the basket.

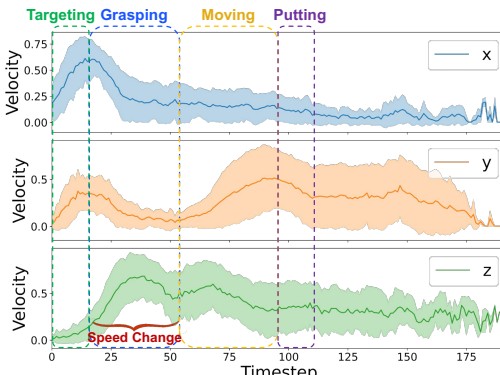

(b) **LIBERO-Object**: Pick up the cream cheese and place it in the basket.

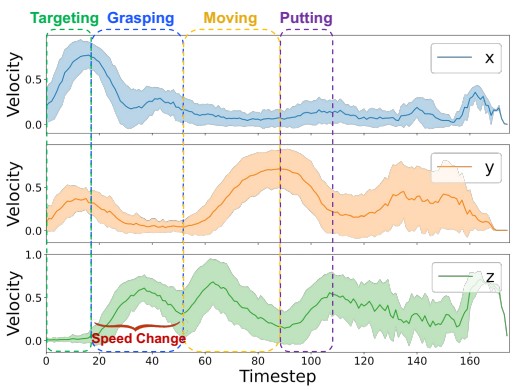

(c) **LIBERO-Object**: Pick up the salad dressing and place it in the basket.

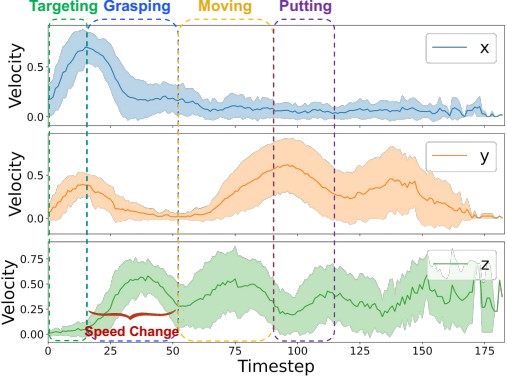

(d) **LIBERO-Object**: Pick up the bbq sauce and place it in the basket.

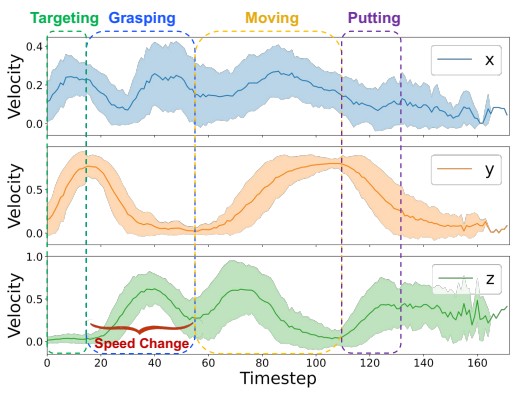

(e) **LIBERO-Object**: Pick up the ketchup and place it in the basket.

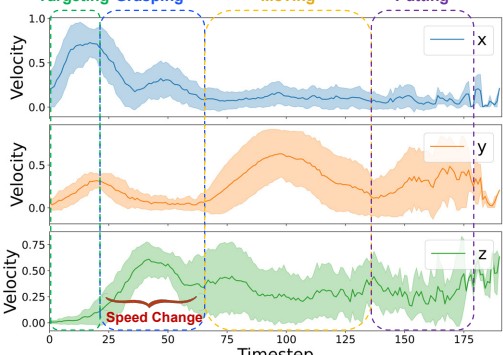

(f) **LIBERO-Object**: Pick up the tomato sauce and place it in the basket.

Figure 14: **Visualizations of SP-VLA on the first 6 LIBERO-Object tasks.**

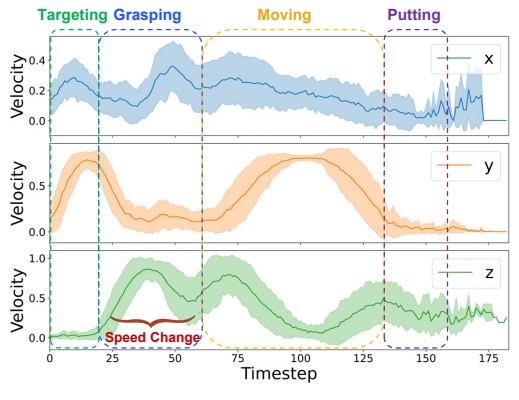

(a) **LIBERO-Object**: Pick up the butter and place it in the basket.

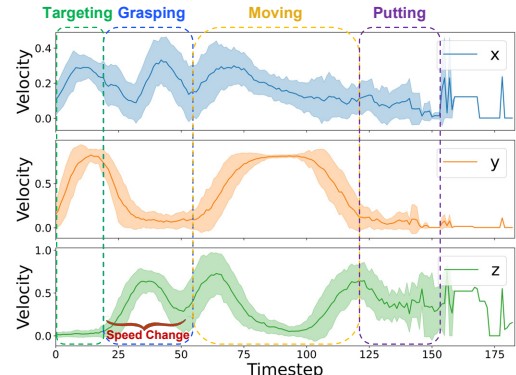

(b) **LIBERO-Object**: Pick up the milk and place it in the basket.

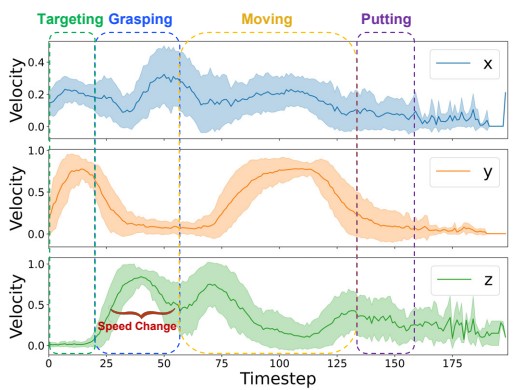

(c) **LIBERO-Object**: Pick up the chocolate pudding and place it in the basket.

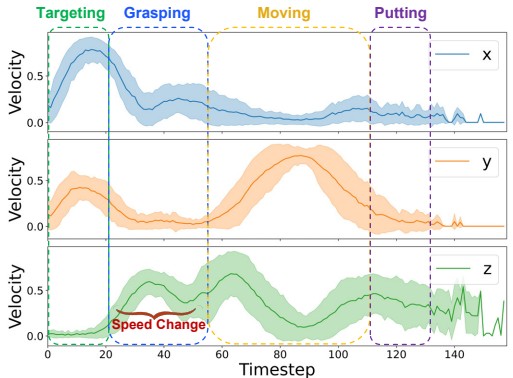

(d) **LIBERO-Object**: Pick up the orange juice and place it in the basket.

Figure 15: **Visualizations of SP-VLA on the remaining 4 LIBERO-Object tasks.**

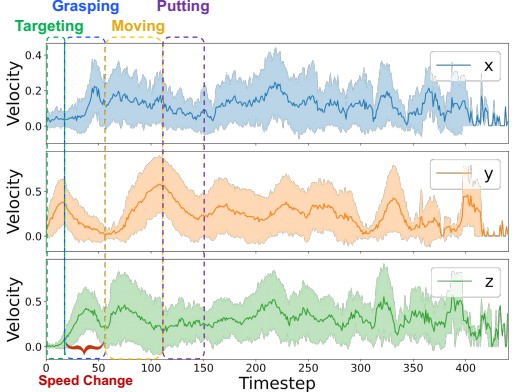

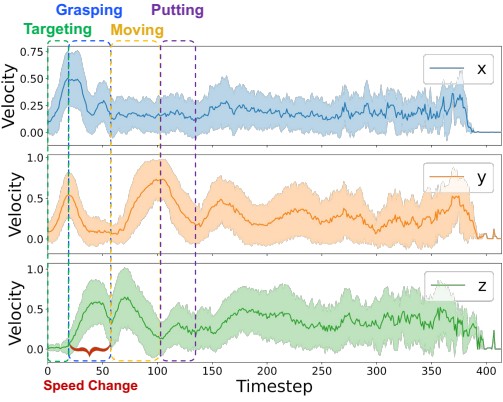

(a) **LIBERO-Long**: Put both the alphabet soup and the tomato sauce in the basket.

(b) **LIBERO-Long**: Put both the cream cheese box and the butter in the basket.

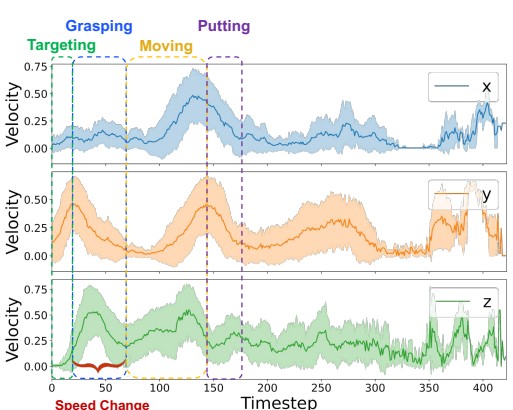

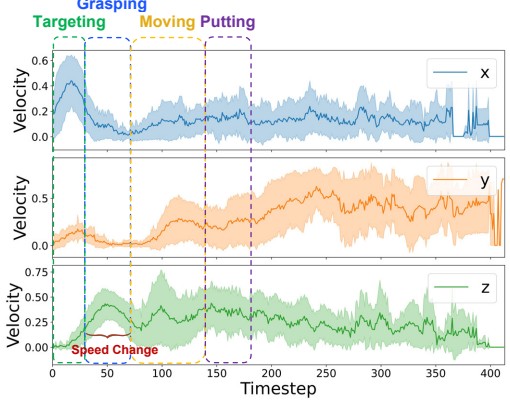

(c) **LIBERO-Long**: Turn on the stove and put the moka pot on it.

(d) **LIBERO-Long**: Put the black bowl in the bottom drawer of the cabinet and close it.

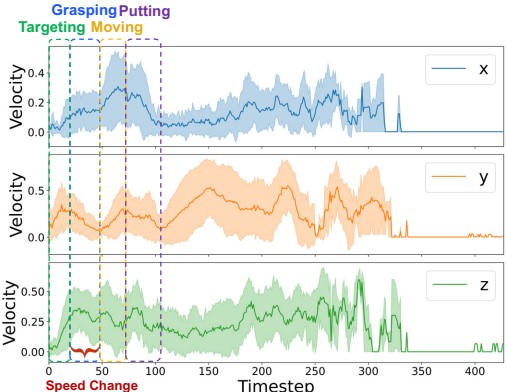

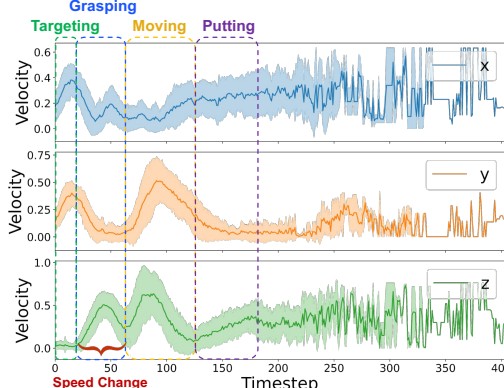

(e) **LIBERO-Long**: Put the white mug on the left plate and put the yellow and white mug on the right plate.

(f) **LIBERO-Long**: Pick up the book and place it in the back compartment of the caddy.

Figure 16: **Visualizations of SP-VLA on the first 6 LIBERO-Long tasks.**

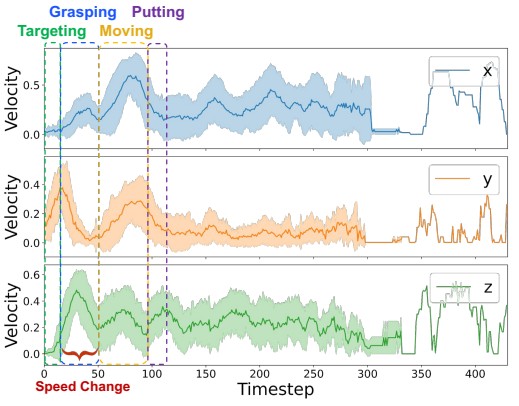

(a) **LIBERO-Long**: Put the white mug on the plate and put the chocolate pudding to the right of the plate.

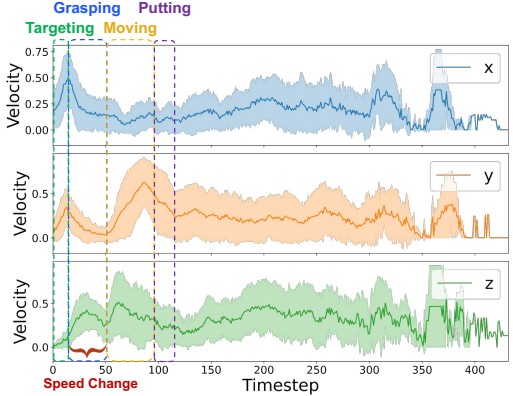

(b) **LIBERO-Long**: Put both the alphabet soup and the cream cheese box in the basket.

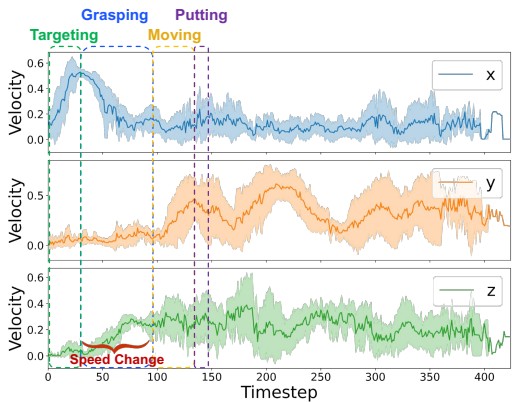

(c) **LIBERO-Long**: Put both moka pots on the stove.

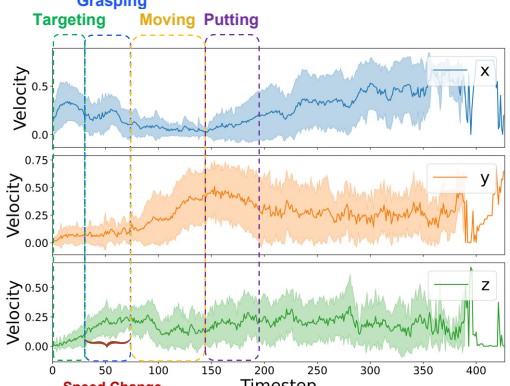

(d) **LIBERO-Long**: Put the yellow and white mug in the microwave and close it.

Figure 17: **Visualizations of SP-VLA on the remaining 4 LIBERO-Long tasks.**

