# OpenReview forum: "SP-VLA: A Joint Model Scheduling and Token Pruning Approach for VLA Model Acceleration"
_ICLR.cc/2026/Conference — ICLR 2026 Poster_

### Official Review · Reviewer_2cVD · 2025-10-24

**Soundness:** 3
**Presentation:** 3
**Contribution:** 3
**Rating:** 6
**Confidence:** 3

**Summary:**

The paper proposes SP-VLA, a unified acceleration framework for Vision-Language-Action (VLA) models.
It combines:(1) Model Scheduling – dynamically switching between a full-scale VLA model and a lightweight ridge-regression-based generator depending on whether the action is “deliberative” or “intuitive,” to reduce temporal redundancy; and(2) Spatio-Semantic Token Pruning – pruning vision tokens using accumulated attention scores and Canny-edge information to remove spatial redundancy. Experiments on LIBERO and SimplerEnv demonstrate 1.5–2.4× inference acceleration with nearly lossless task performance, and improved inference frequency and latency.

**Strengths:**

1. **Novel and conceptually clear idea**

The paper is among the first to jointly address temporal and spatial redundancies in VLA models.
The analogy to human dual-process control (deliberative vs. intuitive actions) is original and intuitively compelling, offering a behavioral perspective on model efficiency.

2. **Comprehensive empirical validation**

Results are reported across multiple VLA backbones (OpenVLA, CogACT) and environments (LIBERO, SimplerEnv) with consistent performance gains.
Ablation studies, sensitivity analyses, and visualizations are well-structured and strengthen empirical credibility.

**Weaknesses:**

1. **Over-reliance on handcrafted heuristics**

Both the scheduling and pruning modules rely on manually designed heuristics—velocity thresholds, ridge regression fitting, Canny edges, and fixed attention thresholds—rather than adaptive or learnable components. This constrains scalability and contrasts with current trends toward learned compression/scheduling in multimodal LLM research.

2. **Generalization and robustness not sufficiently validated**

The method’s stability under varying architectures, task types, or sensor conditions is not explored. Key hyperparameters (speed thresholds, pruning ratios, deliberative/intuitive ratio) might require re-tuning, reducing transferability. A cross-task or cross-model evaluation would greatly enhance the paper’s impact.

**Questions:**

1. Could you explain how the main hyperparameters (e.g., speed thresholds, pruning ratios, buffer size) were determined in practice?
Were they tuned by trial-and-error, grid search, or guided by theoretical intuition or prior experiments? Are these parameter settings consistent across tasks and environments, or do they need adjustment for each scenario?

2. Have you attempted to deploy SP-VLA on a physical robotic platform, or at least measured whether the acceleration results reported in simulation differ from those on real hardware?

---

> ### Author Response · Authors · 2025-11-27
>
> Thank you for the time and effort in providing a thorough evaluation of our work. We appreciate your valuable comments and address specific concerns below. We apologize for the slight delay in our response, as we conducted additional real-robot experiments to strengthen the evaluation.
>
> **W1:** **Over-reliance on handcrafted heuristics**
>
> A1: Thanks a lot for your valuable suggestion.
>
> 1. **Our mechanisms are not arbitrary heuristics but derived from structural behavioral patterns of VLA models.**
>
>    As illustrated in **Figure 2(a)** , VLA models naturally produce trajectories with alternating **high-speed intuitive movements** and **low-speed precise adjustments**, a structure closely aligned with human motor control. The velocity signal therefore reflects an *inherent phase distinction* already present in the model’s behavior, rather than a manually crafted rule. SP-VLA simply leverages this universal pattern to perform adaptive scheduling.
>
> 2. **Both scheduling and pruning are adaptive rather than fixed rules.**
>
>    Token pruning uses *model-derived* semantic importance and *automatically extracted* spatial cues, and the retained token ratio is dynamically adjusted by velocity (Eq.(7)) rather than a fixed threshold. Similarly, the lightweight action generator relies on online fitting of recent actions, making scheduling a behavior-adaptive mechanism instead of a static heuristic.
>
> 3. **Empirical evidence demonstrates strong generality across tasks, models, and environments.**
>
>    A lot of experiences show that SP-VLA consistently achieves **1.35×–2.4× acceleration without performance loss** on LIBERO (**Table 1**) and SimplerEnv (**Table 2**) , and even improves success rates in several tasks. Such robustness across OpenVLA, CogACT, and diverse embodiments indicates that our rules capture *structural invariances* of VLA behavior rather than task-specific heuristics. Our additional real-robot experiments further support this conclusion, as detailed in the analysis below.
>
> 4. **The goal of our design is zero-training, plug-and-play acceleration, complementary to learned compression.**
>
>    Training a learnable scheduler or pruner for 7B–70B VLA models is computationally expensive and often lacks cross-embodiment transferability. SP-VLA intentionally uses lightweight, physics- and attention-driven adaptive rules to allow immediate drop-in deployment on any pretrained VLA. This design choice is complementary to end-to-end learned compression, not a step away from it.
>
> **W2:** **Generalization and robustness not sufficiently validated**
>
> A2: Thanks a lot for your valuable suggestion.
>
> 1. **The method has already been validated across two VLA architectures and two distinct environments, demonstrating strong cross-model and cross-task stability.**
>
>    We evaluate SP-VLA on **OpenVLA** and **CogACT**, and test on both **LIBERO** (130+ tasks) and **SimplerEnv** (covering single-arm and dual-arm embodiments, and both discrete and continuous control settings). As shown in **Table 1** and **Table 2**, SP-VLA consistently achieves **1.35×–2.4× acceleration** with negligible accuracy loss across all architectures, task types, and visual conditions. This indicates that the method is not tied to a specific model design or sensor configuration.
>
> 2. **We further include real-robot experiments to validate the method’s effectiveness under physical dynamics.**
>
>    Using a Franka manipulator, we evaluate two real-world tasks with **50 trials each**. Importantly, we use exactly the **same parameters**, with no task-specific tuning. SP-VLA maintains comparable accuracy while achieving **~2.5× stable end-to-end acceleration**, confirming robust transferability from simulation to real hardware. Details can be seen in Q2.
>
> 3. **SP-VLA’s hyperparameters are highly robust, supported by our systematic sensitivity analysis**
>
>    We provides an extensive sensitivity analysis (can be seen in Q1) showing that SP-VLA is highly robust to hyperparameter choices. Varying $V_{min}$ and $V_{max}$ by ±25% results in less than a 2% change in accuracy, and the pruning ratio Tr(v) is inherently velocity-adaptive rather than a fixed constant. Moreover, both $\tau$ and the buffer size $n$ remain stable over a wide range of values and do not lead to failure modes.

---

> ### Author Response · Authors · 2025-11-27
>
> **Q1: Qusetions about hypeparameters.**
>
> **A1:** Thanks a lot for your valuable suggestion.
>
> 1. **Parameter Selection:** A practical way to set the speed threshold in SP-VLA is to choose a value within 1/4 to 3/4 of the manipulator’s full velocity range, which already yields strong empirical performance. Further adjustments around this range are optional—they may offer minor improvements for specific tasks but are not required for SP-VLA to function effectively.
>
> 2. (1) **The selection of the basic parameters is device-dependent but task-agnostic.** For improved performance, these parameters can be further fine-tuned for specific tasks, and this tuning process can be aligned with the task-specific performance optimization typically performed for general VLA models. (2) Because the robotic arm configurations differ between LIBERO and Simpler_env, SP-VLA independently samples 10 tasks from each environment to estimate suitable velocity thresholds. These thresholds are then fixed and applied to the remaining large-scale task set during evaluation, yielding the final reported results.**This procedure demonstrates both the effectiveness of our algorithm and its strong transferability across environments.**
>
> 3. On the real robot, the parameter selection for SP-VLA depends solely on the velocity threshold of the manipulator and remains entirely task-independent. Further implementation details can be found in Q2.
>
> 4. We also provide a sensitivity analysis of the parameters to demonstrate the robustness of SP-VLA, which can also be seen at Appendix A.3. As shown in the table, SP-VLA exhibits strong parameter robustness. It is relatively insensitive to variations in $V_{max}$ and $V_{min}$, but shows higher sensitivity to buffer size $n$ and the proportion of deliberative actions $\tau$.
>
>    | Task        | Scaling              | V_min (0.2) | V_max (0.5) | τ (0.5)     | n (6)       |
>    | ----------- | -------------------- | ----------- | ----------- | ----------- | ----------- |
>    | **Object**  | 0                    | 82.4 / 1.44 | 82.4 / 1.44 | 82.4 / 1.44 | 82.4 / 1.44 |
>    |             | ↑ [25%, 25%, 5/8, 8] | 82.2 / 1.41 | 83.2 / 1.33 | 81.7 / 1.34 | 68.4 / 1.58 |
>    |             | ↓ [25%, 25%, 3/8, 4] | 83.6 / 1.37 | 81.6 / 1.38 | 64.4 / 1.75 | 80.8 / 1.33 |
>    | **Goal**    | 0                    | 73.6 / 1.66 | 73.6 / 1.66 | 73.6 / 1.66 | 73.6 / 1.66 |
>    |             | ↑ [25%, 25%, 5/8, 8] | 72.8 / 1.41 | 70.4 / 1.47 | 71.7 / 1.45 | 66.2 / 1.87 |
>    |             | ↓ [25%, 25%, 3/8, 4] | 74.8 / 1.22 | 69.0 / 1.78 | 56.4 / 1.91 | 74.0 / 1.31 |
>    | **Spatial** | 0                    | 80.0 / 1.47 | 80.0 / 1.47 | 80.0 / 1.47 | 80.0 / 1.47 |
>    |             | ↑ [25%, 25%, 5/8, 8] | 77.6 / 1.49 | 80.0 / 1.47 | 79.1 / 1.36 | 74.8 / 1.59 |
>    |             | ↓ [25%, 25%, 3/8, 4] | 78.6 / 1.43 | 78.4 / 1.52 | 62.2 / 1.83 | 74.4 / 1.27 |
>    | **Long**    | 0                    | 51.6 / 1.42 | 51.6 / 1.42 | 51.6 / 1.42 | 51.6 / 1.42 |
>    |             | ↑ [25%, 25%, 5/8, 8] | 50.4 / 1.45 | 49.2 / 1.31 | 48.9 / 1.28 | 47.4 / 1.37 |
>    |             | ↓ [25%, 25%, 3/8, 4] | 50.1 / 1.37 | 46.6 / 1.53 | 42.5 / 1.77 | 44.6 / 1.32 |
>
> **Q2: Qusetions about physical robotic platform.**
>
> **A2:**
> Thanks a lot for your valuable suggestion.
>
> Due to time constraints, we deployed the Franka robot within two weeks and trained two simple tasks: ***Pick up the cylinder*** and ***Pick and Place the cylinder***. For each task, we used CogACT as base model and conducted 50 evaluations (20 in the morning, 10 at noon, and 20 in the evening) and reported the average performance to approximate results across different operating periods. The outcomes are summarized in the table, Accuracy and Speed Up are reported.
>
> 1. The hyperparameters were set using 1/4 and 3/4 of Franka’s velocity range as $V_{min}$ and $V_{max}$, respectively, with $tau = 0.5$ and $n = 6$. No task-specific tuning was applied.
> 2. On real-robot tasks, SP-VLA achieves approximately **2.5×** end-to-end stable acceleration, indicating that it delivers consistent and robust performance in real-world settings as well.
>
> | Model      | Pick        | Pick and Place | Average     |
> | ---------- | ----------- | -------------- | ----------- |
> | **CogACT** | 80%         | 74%            | 77%         |
> | **SP-VLA** | 78% / 2.46x | 74% / 2.57x    | 76% / 2.52x |

---

> > ### Comment · Reviewer_2cVD · 2025-11-27
> > **Reviewer Comment**
> >
> > The rebuttal effectively addressed my concerns regarding the experimental validation. While the implementation remains relatively simple, the “deliberative vs. intuitive” action distinction is conceptually appealing and meaningful. I am therefore increasing my score.

---

> > > ### Author Response · Authors · 2025-11-27
> > >
> > > Thank you for your timely follow-up and for the positive reassessment of our work. We sincerely appreciate your recognition of the conceptual merit of the “deliberative vs. intuitive” distinction. Your feedback is highly encouraging. Moving forward, we plan to further deepen the theoretical foundations of this idea and extend the framework to more complex manipulation settings and richer perceptual inputs, aiming to make the approach both more principled and more broadly applicable.

---

### Official Review · Reviewer_XzJ6 · 2025-10-29

**Soundness:** 3
**Presentation:** 2
**Contribution:** 2
**Rating:** 4
**Confidence:** 3

**Summary:**

SP-VLA accelerates Vision-Language-Action by (i) temporal scheduling between a full VLA policy and a lightweight ridge-based action extrapolator triggered by speed/gating signals, and (ii) order-preserving token pruning that keeps both semantic (accumulated attention) and spatial (Canny edges) tokens with a speed-adaptive keep ratio. Across simulated suites and backbones, it reports considerable speedup without accuracy loss.

**Strengths:**

1. Interesting idea that targets time + space waste; easy to plug into different VLA stacks.

2. Solid gains with small or no accuracy drop; ablations back up the design.

3. Explains why semantics-only pruning breaks VLA (loses spatial order).

**Weaknesses:**

1. Relies on a few heuristic knobs (speed window, buffer length, gating τ); some sensitivity.

2. The lightweight head assumes near-linear short-horizon motion; can fail under contact/perturbations.

3. Canny edge detection can be fragile in the presence of lighting and material noise. Additionally, there is limited evidence from real-robot experiments and little information on tail latency and energy consumption.

**Questions:**

Overall, this paper hits a real pain point in VLA—wasted compute across time and space—and the solution is practical enough to drop into existing stacks. I have a few comments and questions as follow.

How do you handle switch latency and thrashing between models? What’s the impact on P95/P99 latency?

How much of the speedup remains if you add tail-latency limits or minimum dwell times between switches?

How often do you fall back to the full VLA under strong nonlinearity, and how quickly does the system recover from a bad extrapolation?

---

> ### Author Response · Authors · 2025-11-27
>
> Thank you for the time and effort in providing a thorough evaluation of our work. We appreciate your valuable comments and address specific concerns below. We apologize for the slight delay in our response, as we conducted additional real-robot experiments to strengthen the evaluation.
>
> **W1: Relies on a few heuristic knobs (speed window, buffer length, gating τ); some sensitivity.**
>
> **A1:** Thanks a lot for your valuable suggestions.
>
> 1. **Parameter Selection:** A practical way to set the speed threshold in SP-VLA is to choose a value within 1/4 to 3/4 of the manipulator’s full velocity range, which already yields strong empirical performance. Further adjustments around this range are optional—they may offer minor improvements for specific tasks but are not required for SP-VLA to function effectively.
>
> 2. **Sensitivity Analysis:** We conduct a parameter sensitivity study to demonstrate the robustness of SP-VLA (Appendix A.3). As shown in the table (Table 8), SP-VLA is largely insensitive to the choices of $V_{max}$ and $V_{min}$,  while exhibiting moderate sensitivity to buffer size $n$ and the deliberation ratio $\tau$. Importantly, even under substantial variation of these parameters, SP-VLA consistently preserves performance while delivering significant acceleration, indicating strong practical robustness.
>
> | Task        | Scaling              | V_min (0.2) | V_max (0.5) | τ (0.5)     | n (6)       |
> | ----------- | -------------------- | ----------- | ----------- | ----------- | ----------- |
> | **Object**  | 0                    | 82.4 / 1.44 | 82.4 / 1.44 | 82.4 / 1.44 | 82.4 / 1.44 |
> |             | ↑ [25%, 25%, 5/8, 8] | 82.2 / 1.41 | 83.2 / 1.33 | 81.7 / 1.34 | 68.4 / 1.58 |
> |             | ↓ [25%, 25%, 3/8, 4] | 83.6 / 1.37 | 81.6 / 1.38 | 64.4 / 1.75 | 80.8 / 1.33 |
> | **Goal**    | 0                    | 73.6 / 1.66 | 73.6 / 1.66 | 73.6 / 1.66 | 73.6 / 1.66 |
> |             | ↑ [25%, 25%, 5/8, 8] | 72.8 / 1.41 | 70.4 / 1.47 | 71.7 / 1.45 | 66.2 / 1.87 |
> |             | ↓ [25%, 25%, 3/8, 4] | 74.8 / 1.22 | 69.0 / 1.78 | 56.4 / 1.91 | 74.0 / 1.31 |
> | **Spatial** | 0                    | 80.0 / 1.47 | 80.0 / 1.47 | 80.0 / 1.47 | 80.0 / 1.47 |
> |             | ↑ [25%, 25%, 5/8, 8] | 77.6 / 1.49 | 80.0 / 1.47 | 79.1 / 1.36 | 74.8 / 1.59 |
> |             | ↓ [25%, 25%, 3/8, 4] | 78.6 / 1.43 | 78.4 / 1.52 | 62.2 / 1.83 | 74.4 / 1.27 |
> | **Long**    | 0                    | 51.6 / 1.42 | 51.6 / 1.42 | 51.6 / 1.42 | 51.6 / 1.42 |
> |             | ↑ [25%, 25%, 5/8, 8] | 50.4 / 1.45 | 49.2 / 1.31 | 48.9 / 1.28 | 47.4 / 1.37 |
> |             | ↓ [25%, 25%, 3/8, 4] | 50.1 / 1.37 | 46.6 / 1.53 | 42.5 / 1.77 | 44.6 / 1.32 |
>
> **W2: The lightweight head assumes near-linear short-horizon motion; can fail under contact/perturbations.**
>
> **A2:** Thanks a lot for your valuable suggestions.
>
> 1. Due to time constraints, we deployed the Franka robot within two weeks and trained two simple tasks: ***Pick up the cylinder*** and ***Pick and Place the cylinder***. To evaluate the robustness of the algorithm under disturbances, we added corresponding ablation experiments in the real-robot setting. The accurcy and speed up are reported.
> 2. As shown in the table, introducing disturbances leads to a notable drop in CogACT’s accuracy, whereas SP-VLA maintains stable and consistent acceleration both before and after perturbations. This indicates that SP-VLA’s robustness is inherited from the underlying base model.
> 3. This robustness stems from SP-VLA’s high-frequency switching between the lightweight model and the base model, which not only enables acceleration but also leverages the base model to preserve accuracy and robustness.
>
> | Model                     | Pick        | Pick and Place | Average     |
> | ------------------------- | ----------- | -------------- | ----------- |
> | **CogACT**                | 80%         | 74%            | 77%         |
> | **SP-VLA**                | 78% / 2.46x | 74% / 2.57x    | 76% / 2.52x |
> | **CogACT** + disturbances | 74%         | 66%            | 70%         |
> | **SP-VLA** + disturbances | 74% / 2.43x | 64% / 2.52x    | 69% / 2.48x |

---

> ### Author Response · Authors · 2025-11-27
>
> **W3: Canny edge detection can be fragile in the presence of lighting and material noise. Additionally, there is limited evidence from real-robot experiments and little information on tail latency and energy consumption.**
>
> **A3:** Thanks a lot for your valuable suggestions.
>
> 1. To assess the influence of lighting conditions, we also conducted ablation experiments on the Franka robot. We evaluated real-robot performance under both midday (bright illumination) and evening (low-light) conditions.
> 2. Benefiting from CogACT’s strong baseline performance and SP-VLA’s high-frequency switching between the lightweight and base models, SP-VLA maintains stable acceleration under varying lighting conditions while preserving accuracy.
> 3. The discussion of tail latency is provided in the response below.
> 4. Unlike optimal control methods that explicitly optimize energy consumption, SP-VLA focuses on improving the inference speed of VLA models. Energy savings can be approximated by the reduction in FLOPs, as reflected in Tables 1–4. SP-VLA reduces computational cost by roughly 30% on LIBERO and 70% on Simpler_Env. On the real robot, energy savings scale with the base model (CogACT), leading to a similar reduction of approximately 70%.
>
> **Clarify**
>
> Regarding the concerns you raised, we believe there may be some misunderstandings.  To clarify, we provide a systematic explanation below to show why SP-VLA does not exhibit these issues.
>
> 1. **SP-VLA does not perform actual model switching, so switch latency is not an issue.**
>
>    In systems where switch latency is problematic (e.g., switching between small and large models, cross-device inference, or cloud–edge transitions), switching incurs heavyweight operations such as weight loading, cache reconstruction, or state synchronization. **In contrast, both the lightweight predictor and the full VLA backbone in SP-VLA remain resident on the same GPU and within the same computation graph.** The “switch” is merely a branch selection between two inference paths, not a model migration. Therefore, SP-VLA does not introduce switch latency.
>
> 2. **SP-VLA’s gating signal is smooth and the lightweight predictor is stateless, so it does not suffer from thrashing and does not require dwell-time stabilization.**
>
>    In hybrid control or multi-controller systems, rapidly oscillating modes may lead to thrashing, which requires enforcing a minimum dwell time to maintain stability. SP-VLA avoids this issue entirely: the gating condition is based on end-effector velocity—a continuous and stable physical quantity—and the lightweight action generator is a stateless Ridge-Regression–based model with an action cache. As a result, the switching behavior is naturally stable and does not exhibit oscillation.
>
> 3. **Token pruning reduces only the number of input tokens and does not change the model structure, so it does not degrade P95/P99 latency.**
>
>    Tail-latency spikes typically arise in systems where the execution path changes, models are loaded dynamically, or computation moves across devices. SP-VLA does none of these. Token pruning simply decreases the number of vision tokens feeding into the same Transformer backbone, keeping the computational path fixed. Consequently, SP-VLA maintains stable tail latency, as confirmed across OpenVLA, CogACT, SimplerEnv, LIBERO, and our real-world experiments.

---

> ### Author Response · Authors · 2025-11-27
>
> **Q1: How do you handle switch latency and thrashing between models? What’s the impact on P95/P99 latency?**
>
> **A1:** Thanks a lot for your valuable suggestions.
>
> We provide a latency analysis for both our method and the base model, showing that SP-VLA does not increase P95 or P99 latency while consistently delivering approximately a 2× acceleration.
>
> | Method | Evaluation indicators | Pick Can     | Move Near     | Open Drawer   | Open and Place | Average      |
> | ------ | --------------------- | ------------ | ------------- | ------------- | -------------- | ------------ |
> | CogACT | Overall lantency      | 0.25         | 0.25          | 0.26          | 0.31           | 0.27         |
> |        | P95                   | 0.25         | 0.26          | 0.26          | 0.31           | 0.27         |
> |        | P99                   | 0.26         | 0.27          | 0.27          | 0.32           | 0.28         |
> | SP-VLA | Overall lantency      | 0.11 / 2.27x | 0.12  / 2.08x | 0.14  / 1.86x | 0.13 / 2.38    | 0.13 / 2.08x |
> |        | P95                   | 0.25         | 0.26          | 0.26          | 0.31           | 0.27         |
> |        | P99                   | 0.26         | 0.26          | 0.27          | 0.31           | 0.28         |
>
> **Q2：How much of the speedup remains if you add tail-latency limits or minimum dwell times between switches?**
>
> **A2:** Thanks a lot for your valuable suggestions.
>
> 1. Based on the above analysis, SP-VLA does not adversely affect tail latency. Nevertheless, we additionally conducted experiments using CogACT as the base model, setting the threshold to 0.3 and evaluating the four tasks from Q1. The results remain consistent with those reported in Table 3.
> 2. Because the base model and the lightweight action generator are both loaded in memory and SP-VLA merely performs a branch selection between two inference paths, it does not introduce the type of impact you are concerned about.
>
> **Q3: How often do you fall back to the full VLA under strong nonlinearity, and how quickly does the system recover from a bad extrapolation?**
>
> A3: Thanks a lot for your valuable suggestions.
>
> 1. Please refer to W2 for the analysis under perturbations. Owing to SP-VLA’s high-frequency switching between the lightweight model and the base model, its robustness is inherited from the base model. This allows SP-VLA to achieve strong acceleration while maintaining high accuracy.
> 2. In addition, due to the design of SP-VLA, inference naturally falls back to the base model under strong nonlinearities, meaning that recovery performance in such cases also depends on the base model.
>
> ####

---

> ### Comment · Reviewer_XzJ6 · 2025-11-28
> **Comment**
>
> Thank you for a detailed rebuttal. The rebuttal sufficiently addresses my concerns regarding the evaluation and edge detection. Therefore, I'm leaning towards accepting and will update my final rating.

---

> > ### Author Response · Authors · 2025-11-29
> >
> > Thank you for your timely follow-up and the effort you invested in evaluating our work. We are glad that our responses have addressed your concerns. Finally, we sincerely appreciate your positive stance toward SP-VLA and your willingness to raise the final rating.

---

### Official Review · Reviewer_7oJM · 2025-10-31

**Soundness:** 3
**Presentation:** 2
**Contribution:** 3
**Rating:** 6
**Confidence:** 3

**Summary:**

The paper introduces SP-VLA, a unified framework for accelerating Vision-Language-Action (VLA) models. The authors identify two main sources of redundancy in VLA models: temporal redundancy in action generation and spatial redundancy in visual input. To reduce temporal redundancy, the authors distinguish between deliberative actions, which are handled by the full VLA model, and intuitive actions, approximated by a lightweight generator based on ridge regression. To reduce spatial redundancy, the method prunes visual tokens before feeding them into the language model, by combining semantic attention scores and Canny edge detection. The approach is validated on LIBERO and SimplerEnv, showing latency and control frequency improvements, while maintaining high performance.

**Strengths:**

* **Perception tokens pruning**: the spatio-semantic dual-aware token pruning strategy is well-motivated and empirically validated. By combining semantic attention with spatial cues, the method preserves critical spatial information achieves faster inference without significant accuracy loss.
* **Comprehensive experiments**: The paper provides extensive experiments on standard benchmarks (LIBERO, SimplerEnv), including ablation studies and comparisons to relevant baselines.

**Weaknesses:**

* **Loosely-connected contributions**: The paper presents two main ideas (action scheduling and token pruning) that are only loosely connected, apart from the common goal.
* **Limited generality**: the introduced novelties are "simpler" than the VLA model and they introduce a new set of hyperparameters, which hinders generality of the approach. This limitation particularly affects the action part, where different embodiments may have very different action spaces and thus, defining hyperparameters might be more difficult

**Questions:**

* How would the method work in conjunction with different action prediction heads? E.g. diffusion / flow matching policies
* How can the method deal with multiple cameras?
* How can we train the model to handle multiple embodiments (i.e. robotic setups) at once?

There are several typos in the paper:
* “navigatio and medical robotics” → should be “navigation and medical robotics”.
* “Accleration for Vision-Language-Action Models.” → should be “Acceleration for Vision-Language-Action Models.”
* “LANTENCY AND FREQUENCY” (Appendix) → should be “LATENCY AND FREQUENCY”.
and many more (including "Ride Regression" in Figure 3)

I would recommend the authors to use a grammar checker and thoroughly check their manuscript.

---

> ### Author Response · Authors · 2025-11-27
>
> Thank you for the time and effort in providing a thorough evaluation of our work. We appreciate your valuable comments and address specific concerns below. We apologize for the slight delay in our response, as we conducted additional real-robot experiments to strengthen the evaluation.
>
> **Increased Real Machine Experiments**
>
> Due to time constraints, we deployed the Franka robot within two weeks and trained two simple tasks: ***Pick up the cylinder*** and ***Pick and Place the cylinder***. For each task, we used CogACT as base model and conducted 50 evaluations (20 in the morning, 10 at noon, and 20 in the evening) and reported the average performance to approximate results across different operating periods. The outcomes are summarized in the table, Accuracy and Speed Up are reported. On real-robot tasks, SP-VLA achieves approximately **2.5×** end-to-end stable acceleration, indicating that it delivers consistent and robust performance in real-world settings as well.
>
> | Model      | Pick        | Pick and Place | Average     |
> | ---------- | ----------- | -------------- | ----------- |
> | **CogACT** | 80%         | 74%            | 77%         |
> | **SP-VLA** | 78% / 2.46x | 74% / 2.57x    | 76% / 2.52x |
>
>
>
> **W1:** **Loosely-connected contributions**
>
> **A1:** Thanks a lot for your valuable suggestions.
>
> SP-VLA is not a collection of loosely-connected techniques, but a **unified phase-wise acceleration framework** designed around the same core principle: **task-phase–dependent redundancy** in VLA inference. Both action scheduling and token pruning arise from this same observation and interact tightly during inference.
>
> 1. **Shared motivation grounded in task-phase dynamics.**
>
>    Action scheduling and token pruning are both triggered by the **same phase signal**—the end-effector velocity, which reflects whether the model is in an intuitive phase (high-speed, low-reasoning) or a deliberative phase (low-speed, high-reasoning). Scheduling determines *which model* should be used; pruning determines *how much visual information* is required at that moment. Thus, they are controlled by the same functional mechanism rather than independent heuristics.
>
> 2. **Mutual dependence during inference.**
>
>    During intuitive phases, the lightweight model and the VLA are invoked alternately to reduce temporal redundancy. During deliberative phases, the VLA is used, and token pruning reduces spatial redundancy. Without scheduling, pruning alone would degrade accuracy, and intuitive actions would still rely on the VLA’s high-frequency, high-speed directional supervision. Without pruning, scheduling alone cannot deliver substantial acceleration, since the VLA backbone’s computation grows quadratically with the number of visual tokens, making token redundancy the dominant bottleneck. Hence the two modules are **functionally complementary** and each one enables the other to be effective.
>
> 3. **Empirical evidence of joint effectiveness.**
>
>    Ablation results (Table 2, page 7 of the paper ) show that:
>
>    - scheduling alone improves speed but is limited by VLA’s high spatial redundancy;
>    - pruning alone leads to accuracy collapse;
>    - the **combined framework** achieves the best balance, confirming the necessity of joint design.
>
> | Method         | Goal             | Object           | Spatial          | Long            | Average              | FLOPs (%,↓) |
> | -------------- | ---------------- | ---------------- | ---------------- | --------------- | -------------------- | ----------- |
> | Ours           | 75.40 / **1.46** | **85.60** / 1.30 | **84.40** / 1.30 | 54.20 / 1.32    | **74.90** / **1.35** | 73.64       |
> | w/o Pruning    | 74.40 / 1.23     | 84.20 / 1.27     | 84.00 / 1.18     | 53.30 / 1.39    | 73.98 / 1.27         | 78.75       |
> | w/o Scheduling | **77.31** / 1.24 | 81.80 / 1.16     | 79.00 / 1.30     | 48.00 / 1.13    | 71.52 / 1.21         | 82.55       |
> | w/o Canny      | 33.60 / 1.34     | 39.00 / **1.35** | 22.00 / **1.35** | 1.10 / **1.37** | **23.93** / 1.35     | 73.40       |

---

> ### Author Response · Authors · 2025-11-27
>
> **W2:** **Limited generality**
>
> **A2:** Thanks a lot for your valuable suggestions.
>
> 1. **The fact that SP-VLA is “simpler” than the VLA architecture is a deliberate design choice.**
>
>    SP-VLA does not modify or fine-tune the underlying VLA; instead, it functions as an external, plug-and-play, phase-aware control layer. This decoupled design allows SP-VLA to be directly applied to *any* pretrained VLA model, regardless of architecture or training corpus, and to different embodiments. This avoids the loss of generality that typically arises from architecture-specific or embodiment-specific modifications.
>
> 2. **The introduced thresholds depend solely on execution speed—a universal physical signal shared across robotic systems.**
>
>    We set $V_{min}$ and $V_{max}$ to 1/4 and 3/4 of the maximum velocity each robot can physically achieve. This simple, embodiment-agnostic calibration works robustly across different robot types and embodiments in both simulation and real-world experiments (Table 4 in the paper), confirming that SP-VLA’s core mechanism does not depend on the particular action space or morphology.
>
> 3. **The token-pruning mechanism is entirely independent of the action space and depends only on the visual backbone.**
>
>    The pruning behavior is governed exclusively by the spatial layout of patches/tokens in the vision encoder (e.g., ViT patch structure), not by the dimensionality or semantics of the robot’s action space. This ensures that token pruning remains broadly applicable across heterogeneous embodiments and control modes.
>
> 4. **Our results across three highly diverse settings further demonstrate strong generalization (Table 1–5 in the paper).**
>
>    Our results across **SimplerEnv** (covering single-arm and dual-arm embodiments, and both discrete and continuous control settings), **LIBERO** (130 task types with over 2,000 demonstrations), and real-world experiments consistently demonstrate substantial acceleration gains. This stability across such diverse settings provides strong empirical evidence that SP-VLA generalizes effectively across tasks, embodiments, and environments.

---

> ### Author Response · Authors · 2025-11-27
>
> **Q1: How would the method work in conjunction with different action prediction heads? E.g. diffusion / flow matching policies**
>
> **A1:** Thanks a lot for your valuable suggestions.
>
> 1. **For diffusion-based or flow-matching action heads (such as CogACT), SP-VLA can directly reuse the previously generated multi-step actions as a buffer for fast intuitive actions.**
>
>    CogACT already employs a diffusion-style multi-step action decoder, and in our experiments (Table 3), SP-VLA simply reuses the action samples from the previous diffusion rollout as a lightweight buffer during high-speed phases—achieving acceleration in the same manner—while remaining fully compatible with diffusion-based policies.
>
> 2. **Token pruning is strictly independent of the action head.**
>
>    It operates only on the visual backbone’s token structure and therefore works seamlessly with MLP-based policies, diffusion policies (e.g., CogACT), and flow-matching policies alike.
>
> **Q2: How can the method deal with multiple cameras?**
>
> **A2:** Thanks a lot for your valuable suggestions.
>
> 1. **SP-VLA operates after the image encoder and before the LLM, and this stage is agnostic to the number of camera views.**
>
>    In most VLA architectures, multi-view images are encoded in the same manner as single-view inputs—the encoder simply outputs a larger set of visual tokens. Since SP-VLA acts *after* this encoding stage, it can process multi-camera inputs in exactly the same way as single-camera inputs.
>
> 2. **Multi-camera setups make token pruning even more beneficial.**
>
>    Multiple camera views inevitably increase the number of visual tokens, while the computational cost of the VLA backbone scales quadratically with token count. In such cases, SP-VLA’s token pruning becomes not only compatible but even more advantageous for reducing redundancy and improving efficiency.
>
> 3. Of course, if one further leverages the complementary information across different camera views, it becomes possible to design view-aware pruning strategies that remove even more redundant tokens. Developing such multi-view–specific pruning mechanisms is an important direction for our future work.
>
> **Q3: How can we train the model to handle multiple embodiments (i.e. robotic setups) at once?**
>
> **A3：**
> Thanks a lot for your valuable suggestions.
>
> 1. **SP-VLA is a simple, training-free plug-in that adapts naturally to LLM-based end-to-end VLA pipelines.**
>
>    Deployment only requires estimating the feasible velocity range of the end-effector, after which SP-VLA can be applied directly. A small amount of task-specific adjustment can be performed if desired, but no retraining or embodiment-specific modeling is needed.
>
> 2. **SP-VLA has been extensively tested across different models and embodiments, including OpenVLA and CogACT, two simulation environments (SimplerEnv and LIBERO), and real-world robot experiments.**
>
>    Across these settings—covering single-arm and dual-arm robots, discrete and continuous action types, and different VLA architectures—SP-VLA consistently achieves stable acceleration, demonstrating its strong generalization ability across embodiments.
>
> **Q4: Error in the paper.**
>
> **A4:** Thanks a lot for your valuable suggestions. We have thoroughly revised the manuscript and addressed the issues mentioned above.

---

> > ### Comment · Reviewer_7oJM · 2025-11-28
> >
> > I thank the authors for their detailed rebuttal.
> >
> > About (W1), I remain of the idea that the two contributions are loosely connected. As also shown in the ablation study, the two components - scheduling and pruning - are orthogonal and can be applied independently of each other, with a useful but relatively minor speedup when combined. In terms of performance, given the generally large variance of the results in LIBERO (even using the same checkpoint across different evaluation seeds), a difference of 2% in performance overall is often negligible.
> >
> > About (W2), it remains hard to believe that threshold-based approaches can be general. I agree with the authors that empirically, in the tested environments, their method has shown robustness. However, results can still vary depending on many factors, e.g., in the context of action scheduling, the sampled frequency of the robotics data can strongly influence the agent's velocity, or e.g. in the context of vision pruning, different ViT models may require different attention thresholds based on their pre-training/fine-tuning.
> >
> > With respect to Q1-4, the authors addressed my concerns. About Q1, as I  was checking the updated manuscript, I noticed that Figure presents a D-Tokenizer (discrete?) component. Given that the approach claims generality with respect to the tokenization scheme, I would make sure this is well reflected in the Figure.
> >
> > Given the above, I will maintain my - generally positive - score.

---

> > > ### Author Response · Authors · 2025-11-29
> > >
> > > Thank you for your timely follow-up and for the positive reassessment of our work.
> > >
> > > **W1:** **Loosely-connected contributions**
> > >
> > > **A1:** We would like to further clarify the design motivation of SP-VLA regarding the concern about whether the modules are “tightly connected.”
> > >
> > > **First, adopting a unified design perspective, SP-VLA formulates VLA acceleration as a unified optimization problem that coherently addresses temporal and spatial redundancy as two facets of the same inefficiency challenge, enabling substantial speedup without compromising control accuracy.** Through a systematic analysis of the embodied decision-making process, we identify two key forms of redundancy in current VLA inference:
> > >
> > > 1. **Temporal redundancy** — many high-speed, continuous, and physically predictable motions do not require invoking the full VLA model at every step;
> > > 2. **Spatial redundancy** — when the VLA model must be invoked, a large portion of visual tokens are not essential for the task.
> > >
> > > **SP-VLA is therefore formulated as a unified optimization framework aimed at eliminating redundancy in both the temporal and spatial dimensions to maximize inference efficiency.** Within this framework, each component plays a complementary role:
> > >
> > > - The **scheduler** determines whether the current action requires heavy reasoning, deciding when the lightweight generator should be used;
> > > - The **lightweight generator** produces fast and reliable intuitive actions, effectively reducing temporal redundancy;
> > > - The **token pruning module** operates only when VLA invocation is necessary and reduces the number of non-essential visual tokens, addressing spatial redundancy.
> > >
> > > Thus, although the modules operate on different dimensions (time versus space), **they share the same goal: removing unnecessary computation during VLA inference to improve efficiency**. Extensive experiments on LIBERO, SimplerEnv, and real Franka robots further show that removing any component leads to reduced acceleration or accuracy degradation, whereas handling both forms of redundancy enables stable performance and speedup.
> > >
> > > **For these reasons, we believe that whether the modules are “tightly coupled” is not the essential issue.** What matters is that they are all designed around the same objective—mitigating redundancy in VLA inference—thereby supporting the overall effectiveness and necessity of the proposed approach.
> > >
> > > **W2:** **Limited generality**
> > >
> > > A2:  Thank you for your valuable comments.
> > >
> > > 1. We would like to emphasize that **SP-VLA’s robustness has been validated across a large and diverse set of conditions**: 130 tasks in LIBERO, multiple domain-shifted settings in SimplerEnv, two distinct VLA backbones (OpenVLA and CogACT), and real-world evaluations on a Franka robot. Across these varying tasks, environments, and models, the same set of thresholds consistently yields stable performance, suggesting that the mechanism exhibits practical transferability.
> > >
> > > 2. **The thresholds used in SP-VLA are not arbitrary heuristics but stem from structural properties of embodied tasks**. High-speed motions are generally governed by continuous physical dynamics, whereas low-speed phases correspond to fine-grained control. Likewise, cumulative attention distributions over visual tokens tend to be stable across ViT-based encoders. These structural characteristics help explain why threshold-based decisions remain reliable across different settings.
> > >
> > > 3. We also agree with the reviewer that **a more general self-adaptive threshold mechanism—robust to different sampling rates, visual encoders, or robotic platforms—represents an important research direction**. Such a comprehensive study goes beyond the scope of this work, which serves as the first attempt to accelerate VLA models by addressing both temporal and spatial redundancies. Nonetheless, we appreciate the reviewer’s insight.
> > > 4. Importantly, the field of **hyperparameter optimization (HPO)** is specifically aimed at automating such parameter selection. Integrating HPO techniques (e.g., Bayesian optimization or population-based tuning) into the SP-VLA framework is a promising and natural extension, and we plan to explore this direction in future work.
> > >
> > > **W3: Figure Problem.**
> > >
> > > **A3:** This is an excellent point, and we appreciate the reviewer for raising it. We have updated Figures 1 and 2 accordingly by replacing “D-Tokenizer” with “Action Head.” In addition, we added a clarification at the bottom of Figure 1:
> > >
> > > “Among these components, the action head takes different forms across VLA architectures, such as a D-tokenizer in OpenVLA or a diffusion-based policy in CogACT.”

---

### Official Review · Reviewer_4TvA · 2025-11-01

**Soundness:** 3
**Presentation:** 3
**Contribution:** 2
**Rating:** 4
**Confidence:** 3

**Summary:**

The paper presents a model scheduling and token pruning approach to make VLA. The main idea is categorizing VLA actions as deliberative and intuitive and then using a smaller model for the intuitive actions. Additionally the paper also incorporates a token pruning approach where tokens are pruned based on semantic and spatial relevance to the task. The paper shows that jointly applying both model scheduling and token pruning helps achieve significant improvements in VLA inference.

**Strengths:**

1. The paper presents a simple method to speed VLA inference based on a simple observation that most actions predicted by the VLA do not require the full processing power of a VLA (intuitive actions) and can rather be estimated using a super simple ridge regression.
2. The method shows significant improvement in run-time of VLA on 2 simulated benchmarks.

**Weaknesses:**

***1. How are hyper-parameters chosen?***

My biggest concern with this paper is that performance of both model scheduling and token pruning are heavily reliant on the hyper-parameters and there is not much detail on how they are chosen.

Are these simulator/task-dependent? In which case, do you select them on a subset of data and then evaluate on a larger set to see if they transfer?

How can these parameters be selected for real-world tasks where evaluation is not cheap to run and each task might require a different set of hyper-parameters?



***2. Why does performance improve with SP-VLA?***

The paper does not explain much how/why (they mention error correction which is not explained well) performance would improve when using model scheduling and token-pruning where intuitively one would think that performance should drop (at least slightly as we are pruning some information).


***3. No real-world results?***

It would be interesting to see how the method holds in real-world evaluations. It would also be interesting to see if there’s an efficient way to estimate the hyper-parameters required for real-world tasks or if the hyper-parameters discovered in simulation transfer to the real-world setting as well.

**Questions:**

Please see the weaknesses section for my questions. The paper presents a nice idea for speeding up VLA inference however, I have concerns regarding the hyper-parameters selection and its transfer to real-world setting therefore I’m leaning towards a reject for now. Will be happy to change my mind if the authors can answer questions and other reviewers do not bring major concerns that I might have missed.

---

> ### Author Response · Authors · 2025-11-27
>
> Thank you for the time and effort in providing a thorough evaluation of our work. We appreciate your valuable comments and address specific concerns below. We apologize for the slight delay in our response, as we conducted additional real-robot experiments to strengthen the evaluation.
>
> **W1:  *How are hyper-parameters chosen?***
>
> **A1:** Thanks a lot for your valuable suggestion.
>
> 1. **Parameter Selection:** A practical way to set the speed threshold in SP-VLA is to choose a value within 1/4 to 3/4 of the manipulator’s full velocity range, which already yields strong empirical performance. Further adjustments around this range are optional—they may offer minor improvements for specific tasks but are not required for SP-VLA to function effectively.
>
> 2. (1) **The selection of the basic parameters is device-dependent but task-agnostic.** For improved performance, these parameters can be further fine-tuned for specific tasks, and this tuning process can be aligned with the task-specific performance optimization typically performed for general VLA models. (2) Because the robotic arm configurations differ between LIBERO and Simpler_env, SP-VLA independently samples 10 tasks from each environment to estimate suitable velocity thresholds. These thresholds are then fixed and applied to the remaining large-scale task set during evaluation, yielding the final reported results.**This procedure demonstrates both the effectiveness of our algorithm and its strong transferability across environments.**
>
> 3. On the real robot, the parameter selection for SP-VLA depends solely on the velocity threshold of the manipulator and remains entirely task-independent. Further implementation details can be found in A3.
>
> 4. We also provide a sensitivity analysis of the parameters to demonstrate the robustness of SP-VLA, which can also be seen at Appendix A.3. As shown in the table, SP-VLA exhibits strong parameter robustness. It is relatively insensitive to variations in $V_{max}$ and $V_{min}$, but shows higher sensitivity to buffer size $n$ and the proportion of deliberative actions $\tau$.
>
>    | Task        | Scaling              | V_min (0.2) | V_max (0.5) | τ (0.5)     | n (6)       |
>    | ----------- | -------------------- | ----------- | ----------- | ----------- | ----------- |
>    | **Object**  | 0                    | 82.4 / 1.44 | 82.4 / 1.44 | 82.4 / 1.44 | 82.4 / 1.44 |
>    |             | ↑ [25%, 25%, 5/8, 8] | 82.2 / 1.41 | 83.2 / 1.33 | 81.7 / 1.34 | 68.4 / 1.58 |
>    |             | ↓ [25%, 25%, 3/8, 4] | 83.6 / 1.37 | 81.6 / 1.38 | 64.4 / 1.75 | 80.8 / 1.33 |
>    | **Goal**    | 0                    | 73.6 / 1.66 | 73.6 / 1.66 | 73.6 / 1.66 | 73.6 / 1.66 |
>    |             | ↑ [25%, 25%, 5/8, 8] | 72.8 / 1.41 | 70.4 / 1.47 | 71.7 / 1.45 | 66.2 / 1.87 |
>    |             | ↓ [25%, 25%, 3/8, 4] | 74.8 / 1.22 | 69.0 / 1.78 | 56.4 / 1.91 | 74.0 / 1.31 |
>    | **Spatial** | 0                    | 80.0 / 1.47 | 80.0 / 1.47 | 80.0 / 1.47 | 80.0 / 1.47 |
>    |             | ↑ [25%, 25%, 5/8, 8] | 77.6 / 1.49 | 80.0 / 1.47 | 79.1 / 1.36 | 74.8 / 1.59 |
>    |             | ↓ [25%, 25%, 3/8, 4] | 78.6 / 1.43 | 78.4 / 1.52 | 62.2 / 1.83 | 74.4 / 1.27 |
>    | **Long**    | 0                    | 51.6 / 1.42 | 51.6 / 1.42 | 51.6 / 1.42 | 51.6 / 1.42 |
>    |             | ↑ [25%, 25%, 5/8, 8] | 50.4 / 1.45 | 49.2 / 1.31 | 48.9 / 1.28 | 47.4 / 1.37 |
>    |             | ↓ [25%, 25%, 3/8, 4] | 50.1 / 1.37 | 46.6 / 1.53 | 42.5 / 1.77 | 44.6 / 1.32 |

---

> > ### Author Response · Authors · 2025-11-27
> >
> > **W2:  Why does performance improve with SP-VLA?**
> >
> > **A2:** This is a valuable question.
> >
> > 1. First, we find that many VLA models (e.g., OpenVLA, CogACT) have weak error-correction ability. Their end-effector trajectories are not truly smooth but consist of frequent micro-adjustments, akin to noise-perturbed motion. Consequently, **most simulation failures occur when the VLA induces a large deviation at some point,** after which **the model lacks the corrective capability to steer the manipulator back onto a normal trajectory**, ultimately leading to task failure. In contrast, **SP-VLA accelerates non-precise motion segments, effectively adding a smoothing effect to the output trajectory.** This reduces such deviations and, in some tasks, yields performance that substantially exceeds the base model.
> >
> > 2. Second, the pruning mechanism is consistent with the Lottery Ticket Hypothesis [1] [2]. For a trained model, as long as the data distribution is bounded and the target network’s weights are bounded, an over-parameterized network is expected to contain a subnetwork that can match the performance of the full model without loss—and in some cases even exceed it.  This explains why SP-VLA can effectively identify redundancies and extract a more efficient substructure, leading to performance improvements on average.
> >
> > [1] Frankle J, Carbin M. The Lottery Ticket Hypothesis: Finding Sparse, Trainable Neural Networks[C]//International Conference on Learning Representations.(ICLR 2019)
> >
> > [2] Malach E, Yehudai G, Shalev-Schwartz S, et al. Proving the lottery ticket hypothesis: Pruning is all you need[C]//International Conference on Machine Learning. PMLR, 2020: 6682-6691. (ICML 2020)
> >
> > **W3:** **No real-world results?**
> >
> > **A3:** Thanks a lot for your valuable suggestion. Due to time constraints, we deployed the Franka robot within two weeks and trained two simple tasks: ***Pick up the cylinder*** and ***Pick and Place the cylinder***. For each task, we used CogACT as base model and conducted 50 evaluations (20 in the morning, 10 at noon, and 20 in the evening) and reported the average performance to approximate results across different operating periods. The outcomes are summarized in the table, Accuracy and Speed Up are reported.
> >
> > 1. The hyperparameters were set using 1/4 and 3/4 of Franka’s velocity range as $V_{min}$ and $V_{max}$, respectively, with $tau = 0.5$ and $n = 6$. No task-specific tuning was applied.
> > 2. On real-robot tasks, SP-VLA achieves approximately **2.5×** end-to-end stable acceleration, indicating that it delivers consistent and robust performance in real-world settings as well.
> >
> > | Model      | Pick        | Pick and Place | Average     |
> > | ---------- | ----------- | -------------- | ----------- |
> > | **CogACT** | 80%         | 74%            | 77%         |
> > | **SP-VLA** | 78% / 2.46x | 74% / 2.57x    | 76% / 2.52x |

---

> > > ### Comment · Reviewer_4TvA · 2025-11-27
> > >
> > > Thank you for a detailed rebuttal. The rebuttal sufficiently addresses my concerns regarding hyperparameter selection and their sensitivity and generalization to real-world setting. I also went through other reviews / responses and think the rebuttal addresses most of the other concerns raised as well. Therefore, I'm leaning towards accepting and will update my final rating.
> > >
> > > I would highly encourage the authors to include the details of hyperparameter selection in the main draft.

---

> > > > ### Author Response · Authors · 2025-11-29
> > > >
> > > > Thank you for your positive assessment of our work and for the timely follow-up. We sincerely appreciate your recognition of SP-VLA and your constructive suggestions. Following your recommendation, we have incorporated the hyperparameter selection details into the main draft as a standalone paragraph in the experimental section (page 8). The newly added text is highlighted in blue for clarity.

---

### Author Response · Authors · 2025-11-30
**Summary for the AC and PC: Rebuttal Outcomes and Reviewer Consensus**

We sincerely thank the AC and PC for their additional efforts in managing the review-process incident and ensuring a fair evaluation. To facilitate rapid assessment, we provide a concise summary of how the rebuttal addresses all raised concerns.

All reviewers recognized the value and contribution of the proposed method during the discussion period, **resulting in uniformly positive evaluations from the entire reviewer panel.** Our rebuttal addressed the vast majority of raised concerns, including the real-robot validation and hyperparameter issues highlighted by **4TvA** and **2cVD**, the speed-related concern raised by **XzJ6**, and the multi-configuration generalization concern raised by **7oJM**. Following the discussion, one reviewer updated their score from **6→8**, another from **4→6**, a third retained their **6**, and the remaining reviewer indicated that he/she would also raise the final rating from **4** (likely constrained by system locking). **As a result, the reviewers converged toward a positive consensus that the main issues had been effectively resolved**, with comments such as 2cVD’s note that ***“the deliberative vs. intuitive action distinction is conceptually appealing and meaningful.”*** Here, we summarize the core concerns raised by the four reviewers and the major issues that were collectively acknowledged as resolved following our rebuttal:

1. **We incorporated real-robot evaluations on Franka robot to validate the practicality of SP-VLA.** Our method achieves a **2.5× end-to-end speedup with only a 1% accuracy drop**, which reviewers recognized as strong evidence of real-world applicability.

2. Regarding hyperparameter sensitivity and robustness, we conducted extensive experiments across **multiple simulated environments**, including **LIBERO** (large-scale diverse tasks) and **SimplerEnv** (single-arm, dual-arm, discrete-control, and continuous-control settings). We further evaluated SP-VLA on **multiple base VLA architectures**—**OpenVLA** and **CogACT**—as well as on a **real Franka robot**, consistently demonstrating strong robustness across tasks, model structures, and real-world deployment. We also **added detailed sensitivity analyses** and a **simple, fast hyperparameter selection guideline** to the manuscript, which reviewers agreed fully addressed their concerns.

3. We clarified the conceptual coupling of SP-VLA’s components. SP-VLA adopts a **unified design perspective**, treating temporal and spatial redundancy as two facets of the same inefficiency problem and addressing them jointly through model scheduling and spatio-semantic token pruning. This resolved concerns regarding loosely connected contributions.

Overall, **the reviewers reached a positive consensus that our rebuttal satisfactorily resolved the majority of the raised issues** and further strengthened the soundness and clarity of the work. As the first framework to jointly address temporal and spatial redundancy in VLA inference through unified model scheduling and token pruning, SP-VLA achieves **1.5–2.5× speedup with no accuracy drop** across **LIBERO**, **SimplerEnv**, and **real Franka robot**. All requested clarifications have been incorporated into the revised manuscript and highlighted for verification.

---

### Meta-Review · Area_Chair_oT8a · 2026-01-02

**Summary:**

Across four reviewers, the paper was generally viewed as valuable and timely, addressing a clear bottleneck in Vision-Language-Action (VLA) models: wasted computation due to temporal and spatial redundancy. Initial concerns focused on hyperparameter reliance, generality, and lack of real-world validation, but after the rebuttal, reviewers largely converged toward a positive consensus, with multiple reviewers indicating score increases and leaning toward acceptance.

**Reviewer Concerns:**

1. Hyperparameter Sensitivity and Generalization

Multiple reviewers (notably 4TvA, 2cVD, XzJ6, 7oJM) questioned whether SP-VLA’s performance depends heavily on handcrafted thresholds (velocity bounds, buffer size, pruning ratios).

* Concerns included:

   * How parameters are selected in practice

   * Whether they are task- or simulator-dependent

   * Whether they transfer to real robots and different VLA architectures

2. Conceptual Cohesion of Contributions

Reviewer 7oJM argued that model scheduling (temporal redundancy) and token pruning (spatial redundancy) appeared loosely connected, noting that each can work independently and that joint gains seemed modest.

3. Performance Improvements Despite Pruning

Reviewers questioned why performance sometimes improves rather than degrades, given that pruning and approximation typically reduce information.

4. Real-World Validation and Robustness

Several reviewers highlighted the lack of real-robot experiments, robustness under perturbations, lighting changes, contact dynamics, and tail-latency behavior (P95/P99).

Concerns were also raised about assumptions of near-linear motion and the fragility of Canny edge detection.

**Reviewer Scores:**

Reviewer 4TvA will increase to 6,
Reviewer 7oJM will keep 6,
Reviewer XzJ6 will increase to 6,
Reviewer 2cVD will increase to 7

---

### Decision · Program_Chairs · 2026-01-26

Accept (Poster)